# Minimum Entropy Coupling with Bottleneck

**M.Reza Ebrahimi**
University of Toronto
mr.ebrahimi@mail.utoronto.ca

**Jun Chen**
McMaster University
chenjun@mcmaster.ca

**Ashish Khisti**
University of Toronto
akhisti@ece.utoronto.ca

## Abstract

This paper investigates a novel lossy compression framework operating under logarithmic loss, designed to handle situations where the reconstruction distribution diverges from the source distribution. This framework is especially relevant for applications that require joint compression and retrieval, and in scenarios involving distributional shifts due to processing. We show that the proposed formulation extends the classical minimum entropy coupling framework by integrating a bottleneck, allowing for a controlled degree of stochasticity in the coupling. We explore the decomposition of the Minimum Entropy Coupling with Bottleneck (MEC-B) into two distinct optimization problems: Entropy-Bounded Information Maximization (EBIM) for the encoder, and Minimum Entropy Coupling (MEC) for the decoder. Through extensive analysis, we provide a greedy algorithm for EBIM with guaranteed performance, and characterize the optimal solution near functional mappings, yielding significant theoretical insights into the structural complexity of this problem. Furthermore, we illustrate the practical application of MEC-B through experiments in Markov Coding Games (MCGs) under rate limits. These games simulate a communication scenario within a Markov Decision Process, where an agent must transmit a compressed message from a sender to a receiver through its actions. Our experiments highlight the trade-offs between MDP rewards and receiver accuracy across various compression rates, showcasing the efficacy of our method compared to conventional compression baseline.

## 1 Introduction

Consider the following Markov Chain modeling a general lossy compression framework $X \xrightarrow{p_{T|X}} T \xrightarrow{q_{Y|T}} Y$, where the input $X$ with a marginal distribution $p_X$, is encoded by the probabilistic encoder $p$ to generate the code $T$. Subsequently, the probabilistic decoder $q$ reconstructs $Y$ from $T$. The objective is to identify the encoder and decoder that minimize the distortion between $X$ and $Y$, subject to an upper bound constraint on the expected code length $H(T) \leq R$.

It is common to measure the sample-wise distortion via direct comparison of $(x, y)$ pairs through a distortion function $d(\cdot, \cdot)$, and consider the expectation $\mathbb{E}[d(X, Y)]$ as a measure of average distortion. Instead, we propose using the logarithmic loss (log-loss) $H(X|Y)$, or equivalently $I(X; Y)$, as an alternative metric to enforce the distortion constraint. The log-loss distortion measure, commonly employed in learning theory, was first explored within rate-distortion theory by Courtade and Wesel [1] and Courtade and Weissman [2]. This measure is particularly suitable in scenarios where reconstructions can be soft, meaning that the decoder produces a distribution rather than a distorted

sample point [3]. Consequently, the optimization problem is formulated as follows:

$$\min_{p_{T|X},\,q_{Y|T}} \quad H(X|Y)$$
$$\text{s.t.} \quad X \leftrightarrow T \leftrightarrow Y, \tag{1}$$
$$H(T) \leq R,$$
$$P(X) = p_X.$$

It is straightforward to check that the optimal solution of (1) is achieved when $T = Y$, with the identity decoder. To address this issue of decoder collapse, we introduce a constraint on the output marginal distribution, $P(Y)$:

---

**Minimum Entropy Coupling with Bottleneck (MEC-B)**

$$\mathcal{I}_{\text{MEC-B}}(p_X, p_Y, R) = \max_{p_{T|X},\,q_{Y|T}} I(X;Y)$$
$$\text{s.t.} \quad X \leftrightarrow T \leftrightarrow Y,$$
$$H(T) \leq R, \tag{2}$$
$$P(Y) = p_Y,$$
$$P(X) = p_X$$

---

The addition of an output distribution constraint is a practical necessity, as in a lossy compression setup the decoder needs to generate outputs following a desired distribution. For example, in image restoration, the output consists of reconstructed images from the code adhering to a certain distribution, possibly the same as the input distribution.

We explore two special cases of (2), where either the encoder or decoder is bypassed. This allows us to optimize the encoder and decoder separately using these cases. First, consider the case where the bottleneck is removed, meaning the constraint $H(T) \leq R$ is relaxed, or $R \geq H(X)$. In this scenario, $X = T$, and the optimization simplifies to:

---

**Minimum Entropy Coupling (MEC)**

$$\mathcal{I}_{\text{MEC}}(p_X, p_Y) = \max_{p_{Y|X}} I(X;Y)$$
$$\text{s.t.} \quad P(Y) = p_Y, \tag{3}$$
$$P(X) = p_X$$

---

This involves identifying the probabilistic mapping $p_{Y|X}$ between the marginals $p_X$ and $p_Y$ that maximizes the obtained mutual information. This problem, as described in (3), has been extensively studied in the literature as minimum entropy coupling (MEC), with early research conducted by Vidyasagar [4], Painsky et al. [5], Kovačević et al. [6], Cicalese et al. [7], among others. Thus, we define the original problem presented in (2) as minimum entropy coupling with bottleneck (MEC-B). Next, consider the case where the decoder is removed, resulting from the relaxation of the output distribution constraint in (2):

---

**Entropy-Bounded Information Maximization (EBIM)**

$$\mathcal{I}_{\text{EBIM}}(p_X, R) = \max_{p_{T|X}} I(X;T)$$
$$\text{s.t.} \quad H(T) \leq R, \tag{4}$$
$$P(X) = p_X$$

---

Similar to minimum entropy coupling, this problem identifies the joint distribution between two random variables that maximizes their mutual information. However, rather than imposing a marginal distribution constraint, it enforces a more flexible entropy constraint on one of the variables. Lemma 1 provides a decomposition for the mutual information between input and output $I(X;Y)$, given the Markov chain $X \leftrightarrow T \leftrightarrow Y$.

**Lemma 1.** *Given a Markov chain $X \leftrightarrow T \leftrightarrow Y$:*

$$I(X;Y) = I(X;T) + I(Y;T) - I(T;X,Y). \tag{5}$$

The proof follows multiple applications of the chain rule for mutual information. The following lower bound on the MEC-B objective is attainable based on Lemma 1:

$$I(X;Y) \geq I(X;T) + I(Y;T) - R. \tag{6}$$

In this work, we consider maximizing the lower bound (6) as a proxy to the main objective. This allows a decomposition of the encoder and decoder for the MEC-B formulation in (2):

1. **Encoder Optimization:** The encoder is first optimized separately, according to Entropy-Bounded Information Maximization in (4), $\mathcal{I}_{\text{EBIM}}(p_X, R)$, resulting in the marginal distribution $\hat{p}_T$ on the code $T$.

2. **Decoder Optimization:** The decoder is then optimized by solving a minimum entropy coupling in (3) between the code and output marginals, $\mathcal{I}_{\text{MEC}}(\hat{p}_T, p_Y)$.

Therefore, in terms of problems (2), (3), and (4):

$$\mathcal{I}_{\text{MEC-B}}(p_X, p_Y, R) = \max_{p_{T|X}, \, q_{Y|T}} I(X;Y) \tag{7}$$

$$\geq \max_{p_{T|X}, \, q_{Y|T}} \big( I(X;T) + I(Y;T) - R \big) \tag{8}$$

$$\geq \mathcal{I}_{\text{EBIM}}(p_X, R) + \mathcal{I}_{\text{MEC}}(\hat{p}_T, p_Y) - R. \tag{9}$$

In this paper, we address the Entropy-Bounded Information Maximization problem in Section 3, providing theoretical insights into the solution structure across the entire spectrum of rate limits. We establish an upper bound on the objective and demonstrate that only deterministic mappings can achieve this bound. Then, in Section 3.1, we introduce a greedy algorithm designed to identify deterministic mappings with a guaranteed input-dependent gap from the optimal solution. Subsequently, in Section 3.2, we describe a method to identify optimal mappings near any deterministic mapping, effectively bridging the gap between discrete deterministic mappings and providing deeper theoretical insights into the problem structure.

Following this theoretical groundwork, Section 4 applies the MEC-B framework to extend Markov Coding Games (MCG) with communication bottlenecks between the source and the agent. Experimental results for MCGs with rate limits are detailed in Section 4.2, showcasing the practical implications of our theoretical developments. The appendix sections complement these discussions by including formal proofs for all theorems and lemmas, a concise overview of the original minimum entropy coupling problem, and additional experimental results.

## 2 Related Work

**Couplings and Minimum Entropy Coupling** A fundamental problem in probability theory, known as coupling, concerns determining the *optimal* joint distribution of random variables given their marginal distributions. This problem has a long history, with early examples by Fréchet [8] seeking the joint distribution that maximizes correlation subject to marginal constraints. References [9–12] provide a broader treatment of these problem classes and their applications. Notably, optimal transport (OT) emerges as a significant class within this framework, where optimality is defined as minimizing the expected value of a loss function over the joint distribution. See [13] for an in-depth treatment of the optimal transport problem.

The minimum entropy coupling (MEC) focuses on finding the joint distribution with the smallest entropy given the marginal distribution of some random variables. This problem has been first studied in [4–7], among others. While it is shown by Vidyasagar [4], Kovačević et al. [6] that MEC is NP-Hard, the literature contains many approximation algorithms for this problem. One of the earliest greedy algorithms for MEC was introduced by Kocaoglu et al. [14] in the context of causal inference, achieving a local minimum with a gap of $1 + \log n$ bits from the optimum, where $n$ represents the size of the alphabet. This bound was further improved in subsequent works [15, 16].

Based on tools from the theory of majorization [17], Cicalese et al. [18] developed a new greedy algorithm producing solutions 1 bit away from the optimal. Subsequent improvements by Li [19] enabled the construction of a coupling whose entropy is within 2 bits of the optimal value, regardless of the number of random variables involved. Despite these advances, Compton [15] identified a *majorization barrier* that limits further improvements, while Compton et al. [16] introduced the profile method offering stronger lower bounds for the coupling entropy.

Minimum entropy coupling finds innovative applications beyond causal inference [14, 20, 21]. For instance, Sokota et al. [22] utilized it in Markov coding games to enable reinforcement learning agents to communicate via Markov decision process trajectories. This application showcased MEC's utility in enabling efficient information transmission through constrained environments like video game interactions. Similarly, de Witt et al. [23] applied MEC to securely encode secret information in regular text, showing MEC corresponds to the maximally efficient secure procedure.

**Lossy Source Coding** While log-loss is widely used in prediction and learning, its application as a distortion measure in the context of source coding has been less explored, with the earliest examples appearing in [1] and [2]. Log-loss is particularly suited as a distortion measure in soft reconstructions, meaning the decoder outputs a distribution. Shkel and Verdú [3] explored a single-shot lossy source coding setting under logarithmic-loss, using a straightforward encoding scheme. Unlike the EBIM formulation in (4) which imposes a direct entropy constraint on the code, this approach constrained the code by the cardinality of its support.

Finally, Blau and Michaeli [24] introduced the Rate-Distortion-Perception (RDP) tradeoff in lossy compression. The RDP framework does not fix the output distribution; instead, it imposes a softer perceptual constraint on the generated outputs. Additionally, our work incorporates an entropy constraint on $T$ as a rate bottleneck, while in the RDP formulation, $I(X;Y)$ can be interpreted as the rate bottleneck. In this line of research, the work of Liu et al. [25] is closest in spirit to our approach, as the authors studied a lossy compression setting with different source and reconstruction distributions. They demonstrated that their setting could be formulated as a generalization of optimal transport with an entropy bottleneck. However, they used mean squared error (MSE) as the distortion metric, while we consider log-loss. Therefore, the mathematical machinery required for our analysis differs significantly from prior work.

## 3   Entropy-Bounded Information Maximization

Consider a discrete random variable $X$ defined over the alphabet $\mathcal{X} = \{1, \ldots, n\}$ with a given marginal probability distribution $p_X$. The following problem aims to establish a maximal information coupling between $X$ and another random variable $T$, defined over the alphabet $\mathcal{T} = \{1, \ldots, m\}$, where the entropy of $T$ is constrained to be no more than $R$ bits. Unlike minimum entropy coupling, the marginal distribution of the second random variable $T$ is not predetermined; the only constraint on $T$ is its entropy.

$$\mathcal{I}_{\text{EBIM}}(p_X, R) = \max_{p_{XT} \in \mathcal{M}} I(X;T), \tag{10}$$

where set $\mathcal{M}$ consists of all joint distributions $p_{XT}$ that satisfy the following conditions:

1. $\sum_t p_{XT}(x, t) = p_X(x)$, ensuring that the marginal distribution of $X$ is preserved.
2. $H(T) \leq R \leq H(X)$, ensuring the entropy of $T$ is constrained to be no more than $R$.

We call this problem Entropy-Bounded Information Maximization (EBIM). Note that the objective in (10) is upper-bounded by $R$, since:

$$\begin{aligned}
\mathcal{I}_{\text{EBIM}}(p_X, R) &= \max_{p_{XT} \in \mathcal{M}} I(X;T) \\
&\leq \max_{p_{XT} \in \mathcal{M}} H(T) \leq R.
\end{aligned} \tag{11}$$

The following theorem establishes that only deterministic couplings can achieve this upper-bound.

**Theorem 1.** *$\mathcal{I}_{EBIM}(p_X, R) = R$ if and only if there exists a function $g : \mathcal{X} \to \mathcal{T}$ such that $H(g(X)) = R$.*

The formal proof is presented in Section A.2. Note that the mutual information $I(X;T)$ is invariant to permutations on $\mathcal{T}$. Specifically, for any permutation $\pi : \mathcal{T} \to \mathcal{T}$, we have $I(X;T) = I(X, \pi(T))$.

Given that problem (10) only constrains the entropy $H(T)$, the objective is indifferent to such permutations. Let us define the permutation group of a joint distribution $p_{XT}$ as:

$$\mathcal{P}(p_{XT}) = \{P \mid P(x, \pi(t)) = p_{XT}(x,t), \ \forall \pi : \mathcal{T} \to \mathcal{T}\}. \tag{12}$$

**Remark 1.** *Each partition of $\mathcal{X}$ is associated with a permutation group of a deterministic mapping. Consequently, the total number of potential deterministic mapping groups, independent of the entropy constraint on $T$, will be the total number of feasible partitions of $\mathcal{X}$. The total number of ways to partition a set of size $n$ corresponds to the $n$-th Bell number, symbolized by $B_n$. The growth rate of the Bell numbers is $\mathcal{O}(n^n)$ [26], rendering brute force iteration of all deterministic mappings infeasible.*

In Figure 2 (left), a brute force method is applied to solve EBIM (10) for an input alphabet of size three. As observed, there are five potential partitions on $\mathcal{X}$, each corresponding to a point where $\mathcal{I}_{\text{EBIM}}(p_X, R) = R$. Given the impracticality of brute force search for large alphabet sizes, in Section 3.1, we introduce a greedy search algorithm to identify deterministic mappings with a guaranteed performance gap from the optimal. Following this, in Section 3.2, we explore optimal mappings close to these deterministic mappings, providing a strategy to narrow the gap between the identified deterministic mappings.

## 3.1 Proposed Search Algorithm for Deterministic Mappings

Since iterating over all deterministic mappings is not feasible, one should look for carefully constructed search algorithms to find such mappings with resulting $H(T)$ as close as possible to $R$. Without the loss of generality, suppose $p_X = [p_1, p_2, \cdots, p_n]$ is arranged in a decreasing order. Algorithm 1 presents a search approach for discovering a deterministic mapping $T = g(X)$, resulting in $I(X; T)$ that is at most $h(p_2)$ bits away from the optimal $\mathcal{I}_{\text{EBIM}}(p_X, R)$, where $h(\cdot)$ is the binary entropy function: $h(p) = -p \log(p) - (1 - p) \log(1 - p)$.

---
**Algorithm 1** Deterministic EBIM Solver
---
**Input:** $p_X, R$
**Output:** $p_{XT}$
1: $p_{XT} \leftarrow \text{diag}(p_X)$
2: **if** $R \geq H(X)$ **then**
3:      **return** $p_{XT}$
4: **for** $i \leftarrow 1$ to $|p_X| - 1$ **do**
5:      $p_s^{(i)} \leftarrow$ Merge the two columns with the smallest sum in $p_{XT}$.
6:      $I_s^{(i)} \leftarrow$ Mutual Information imposed by $p_s^{(i)}$.
7:      $p_l^{(i)} \leftarrow$ Merge the two columns with the largest sum in $p_{XT}$.
8:      $I_l^{(i)} \leftarrow$ Mutual Information imposed by $p_l^{(i)}$.
9:      **if** $I_s^{(i)} \leq R$ **then**
10:         **return** $p_s^{(i)}$
11:      **else if** $I_l^{(i)} \leq R < I_s^{(i)}$ **then**
12:         **return** $p_l^{(i)}$
13:      **else**
14:         $p_{XT} \leftarrow p_l^{(i)}$
---

The deterministic EBIM solver in Algorithm 1 has $\mathcal{O}(n \log n)$ time complexity, where $n$ is the cardinality of the input alphabet. This is because the main loop of the algorithm runs for at most $n$ steps (as at each step we combine two elements of the input distribution) and finding min/max elements can be done in $\mathcal{O}(\log n)$ using a heap data structure. Also, the mutual information calculation at each step can be done in constant time by only calculating the decrease in entropy after combining two elements of the distribution.

**Theorem 2.** *If the output of Algorithm 1 yields mutual information $\widehat{I}$, then*

$$\mathcal{I}_{\text{EBIM}}(p_X, R) - \widehat{I} \leq h(p_2), \tag{13}$$

*where $h(\cdot)$ is the binary entropy function, and $p_2$ denotes the second largest element of $p_X$.*

Let $n = |p_X|$. The procedure outlined in Algorithm 1 establishes a series of deterministic mappings $p_s^{(1)}, p_l^{(1)}, \cdots, p_s^{(n-1)}, p_l^{(n-1)}$, corresponding to a decreasing sequence of mutual information values $I_s^{(1)}, I_l^{(1)}, \cdots, I_s^{(n-1)}, I_l^{(n-1)}$. The algorithm then picks the mapping with the highest mutual information that does not exceed $R$. The proof involves establishing an upper bound on the gap between these successive mutual information values. A formal proof is presented in Section A.3.

It is important to highlight that the gap described in Theorem 2 is bounded by one bit, i.e., $h(p_2) \leq 1$, with equality achieved when $p_X = [0.5, 0.5]$. Although the gap is capped at one bit, the maximal mutual information in the EBIM formulation scales with $R$. Thus, the most natural interpretation of this gap emerges in higher rate regimes. Furthermore, the gap described in Theorem 2 remains small when $p_2$ is small, specifically in cases where the input alphabet size is large and the distribution is not heavily skewed toward a few elements.

## 3.2 Optimal Coupling Around Deterministic Mappings

Section 3.1 introduced a greedy search algorithm designed to identify deterministic mappings with a guaranteed and input-dependent gap from the optimal. In this section, we find the optimal couplings close to any deterministic mapping. This method allows us to close the gap between the mappings identified by Algorithm 1, as will be demonstrated later.

**Theorem 3.** *Let $p_{XT}$ denoted by a $|\mathcal{X}| \times |\mathcal{T}|$ matrix, defines a deterministic mapping $T = g(X)$, with $I(X;T) = H(T) = R_g$. We have $\mathcal{I}_{EBIM}(p_X, R_g) = R_g$, and for small enough $\epsilon > 0$:*

1. *$\mathcal{I}_{EBIM}(p_X, R_g + \epsilon)$ is attained as follows:*
   *Normalize the columns by dividing each column by its sum. Then, select the cell with the smallest normalized value and move an infinitesimal probability mass from this cell to a new column of $p_{XT}$ in the same row.*

2. *$\mathcal{I}_{EBIM}(p_X, R_g - \epsilon)$ is achieved as follows:*
   *Identify the columns with the smallest and largest sums in $p_{XT}$. Select the cell with the smallest value in the column with the lowest sum. Transfer an infinitesimal probability mass from this cell to the column with the highest sum in the same row.*

Figure 1 on the right depicts an example of optimal solutions in the neighborhood of a deterministic mapping. While Algorithm 1 effectively identifies deterministic mappings that produce mutual information close to the budget $R$, Theorem 3 can help bridge the remaining gap. More specifically, one can begin with a deterministic mapping and use two probability mass transformations outlined in Theorem 3 to navigate across the $I$-$R$ plane.

Figure 2 illustrates this strategy; for $p_X = [0.7,\ 0.2,\ 0.1]$, identifying all 5 possible deterministic mappings is straightforward. Applying the transformations from Theorem 3 then yields various solutions across the $I$-$R$ plane (represented by dashed lines). Subsequently, one can select the solution that maximizes mutual information for any given value of $R$ (highlighted with a thick solid line),

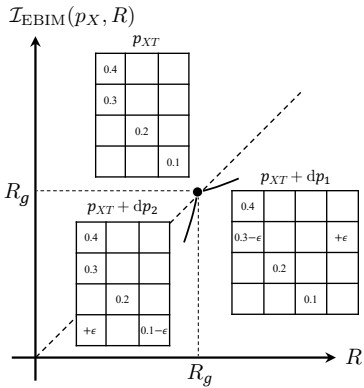

Figure 1: An example for Theorem 3.

thus producing a comprehensive solution for every value of $R$. As demonstrated in Figure 2, this strategy recovers the optimal solutions, as determined by brute force, for the simple case of an input alphabet of three. However, while effective, the optimality of this approach remains a conjecture.

## 4  Application: Markov Coding Game with Rate Limit

Markov Coding Games (MCGs), as introduced by Sokota et al. [22], represent a specialized type of multi-player decentralized Markov Decision Processes (MDPs) involving several key components: a source, an agent (sender), a Markov decision process, and a receiver. An MCG episode unfolds in three stages: initially, the agent receives a private message from the source, which it must then indirectly convey to the receiver. Next, the agent participates in an episode of the Markov decision

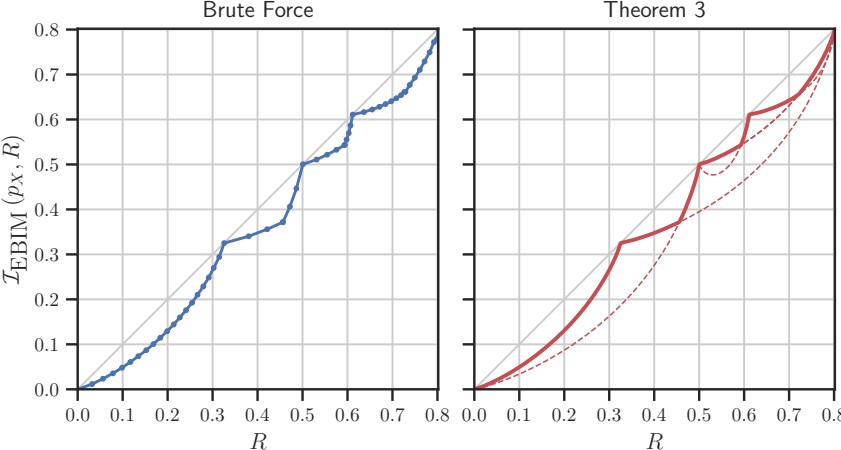

Figure 2: Solutions to the EBIM problem for $p_X = [0.7, 0.2, 0.1]$. **Left:** brute force solution. **Right:** application of the transformations from Theorem 3 to each deterministic mapping (dashed lines) and selection of solutions with maximal mutual information for each $R$ value (thick solid line). This strategy effectively recovers optimal solutions, aligning with those found by brute force in this case.

process. Finally, the receiver attempts to decode the original message based on the observed MDP trajectory. The overall reward is a combination of the MDP payoff and the accuracy with which the receiver decodes the message. MCGs are particularly interesting due to their ability to generalize frameworks like referential games [27] and source coding [28].

We will consider a natural extension to Markov Coding Games, where the link from the source to the agent is rate-limited. This means, contrary to the original setting of Sokota et al. [22], the agent does not fully observe the message at each MDP round, but will receive a compressed version of the message iteratively, and in turn, encodes information about the message in the MDP trajectory for the receiver.

Following Sokota et al. [22], we define a rate-limited MCG as a tuple $\langle (\mathcal{S}, \mathcal{A}, \mathcal{T}, \mathcal{R}), \mathcal{M}, \mu, \zeta, R \rangle$, where $(\mathcal{S}, \mathcal{A}, \mathcal{T}, \mathcal{R})$ is an MDP denoted by state and action spaces, and reward and transition functions, respectively. $\mathcal{M}$ is a set of messages, $\mu$ is the prior distribution over messages $\mathcal{M}$, $\zeta$ is a non-negative real number we call the message priority, and finally, $R$ is the communication rate limit between the source and the agent. An MCG episode proceeds in the following steps:

1. Message $M \sim \mu$ is sampled from the prior over messages at the source.
2. Based on the selected message $M$ and the history of the MDP episode, the source generates and transmits signal $T$ to the agent, adhering to the rate limit $R$.
3. The Agent uses a conditional policy $\pi_{|T}$, which takes current state $s \in \mathcal{S}$ and received signal $T$ as input and outputs distributions over MDP actions $\mathcal{A}$, to generate the next action $a$.
4. After repeating steps 2 and 3, the agent's terminal MDP trajectory $Z$ is given to the receiver as an observation.
5. The receiver uses the terminal MDP trajectory $Z$ to output a distribution over messages $\mathcal{M}$, estimating the decoded message $\hat{M}$.

The objective of the agents is to maximize the expected weighted sum of the MDP reward and the accuracy of the receiver's estimate $\mathbb{E}[\mathcal{R}(Z) + \zeta \mathbb{I}[M = \hat{M}]]$ [22]. Optionally, If a suitable distance function exists, instead, the objective can also be adjusted to minimize the difference between the actual message and the guess. A diagram of the structure MCG with rate limit is shown in Figure 3.

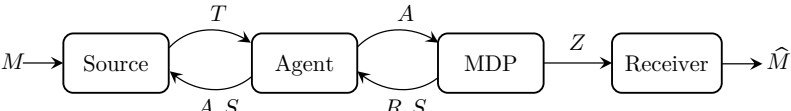

Figure 3: The structure of a Markov Coding Game with Rate Limit.

## 4.1 Method Description

**Marginal Policy** Following Sokota et al. [22], before execution we first derive a marginal policy $\pi$ for the MDP, based on the Maximum-Entropy reinforcement learning objective:

$$\max_\pi \mathbb{E}_\pi \left[ \sum_t R(S_t, A_t) + \beta H(A_t|S_t) \right]. \quad (14)$$

The value of $\beta$ in (14) needs to be determined in accordance with the message priority $\zeta$ of the MCG. Note that this marginal policy does not depend on the choice of message. By introducing stochasticity into this policy, we can encode information about the message into the selection of actions at each step during runtime. For more details, see Section C.1.

**Step 1 - Source: Message Compression** At the beginning of each round, given the updated message belief $p_M$, the source compresses the message to generate the signal $T$, adhering to the source-agent rate limit $R$, by solving EBIM in (10). The source then transmits the signal $T$ to the agent. Subsequently, after observing the action taken by the agent, the source updates the message belief for the next round. Algorithm 2 outlines the steps taken by the source.

**Step 2 - Agent: Minimum Entropy Coupling** As illustrated in Algorithm 3, at each round, upon receiving the signal $T$, the agent constructs a conditional policy $\pi_{|T}$ by performing minimum entropy coupling between the action distribution from the marginal policy $\pi(s)$ with the signal distribution $p_T$. Subsequently, the next action is sampled from the conditional policy, $a \sim \pi_{|T}$. Finally, the agent updates the message belief based on the chosen action.

**Receiver: Decoding the Message** Given the agent's final MDP trajectory, the receiver mirrors the actions of the source and agent to update the message belief at each step. As outlined in Algorithm 4, the process begins with the receiver compressing the message based on the current message belief. This is followed by performing minimum entropy coupling between the marginal policy and the distribution of the compressed message. The final message belief is used to estimate the decoded message.

---

**Algorithm 2** Source

1: **Input**: $\pi, \mu, R, s^0$
2: Observe message $m \sim \mu$
3: Initialize message belief $p_M \leftarrow \mu$
4: Initialize state $s \leftarrow s^0$
5: **while** Source's turn **do**
6: $\quad p_{MT} \leftarrow \texttt{compress}(p_M, R)$
7: $\quad t \sim p_{T|M}(m)$
8: $\quad$ **Send** $t$ to the agent.
9: $\quad p_T \leftarrow \sum_{m'} p_{MT}(m', \cdot)$
10: $\quad p_{TA} \leftarrow \texttt{min\_ent\_coupling}(p_T, \pi(s))$
11: $\quad p_{MA} \leftarrow \sum_{t'} p_{MT}(\cdot, t')\, p_{A|T}(t')$
12: $\quad a, s \leftarrow$ **Observe** action and next state
13: $\quad p_M \leftarrow p_{M|A}(a)$

---

**Algorithm 3** Agent

1: **Input**: $\pi, \mu, R, s^0$
2: Initialize message belief $p_M \leftarrow \mu$
3: Initialize state $s \leftarrow s^0$
4: **while** Agent's turn **do**
5: $\quad p_{MT} \leftarrow \texttt{compress}(p_M, R)$
6: $\quad$ **Receive** $t$ from the source.
7: $\quad p_T \leftarrow \sum_{m'} p_{MT}(m', \cdot)$
8: $\quad p_{TA} \leftarrow \texttt{min\_ent\_coupling}(p_T, \pi(s))$
9: $\quad \pi_{|T} \leftarrow p_{A|T}(t)$
10: $\quad a \sim \pi_{|T}$
11: $\quad s \leftarrow$ Commit action $a$ to MDP.
12: $\quad p_{MA} \leftarrow \sum_{t'} p_{MT}(\cdot, t')\, p_{A|T}(t')$
13: $\quad p_M \leftarrow p_{M|A}(a)$

---

**Algorithm 4** Receiver

1: **Input**: $z, \pi, \mu, R, s^0$
2: Initialize message belief $p_M \leftarrow \mu$
3: Initialize state $s \leftarrow s^0$
4: **for** $s, a \in z$ **do**
5: $\quad p_{MT} \leftarrow \texttt{compress}(p_M, R)$
6: $\quad p_T \leftarrow \sum_{m'} p_{MT}(m', \cdot)$
7: $\quad p_{TA} \leftarrow \texttt{min\_ent\_coupling}(p_T, \pi(s))$
8: $\quad p_{MA} \leftarrow \sum_{t'} p_{MT}(\cdot, t')\, p_{A|T}(t')$
9: $\quad p_M \leftarrow p_{M|A}(a)$
10: **return** $\arg\max_{m'} p_M(m')$

---

## 4.2 Experimental Results

This section presents the experimental results of the method described in Section 4.1, applied to Markov Coding Games. For our experiments, we utilize a noisy *Grid World* environment for the Markov Decision Process. Section C.2 provides more detail on the environment setup used in this experiment.

The marginal policy is learned through Soft Q-Value iteration, as described in Algorithm 8. By increasing the value of $\beta$ in Equation (14), we induce more randomness into the marginal policy. Consequently, higher values of $\beta$ lead to an increase in the total number of steps taken by the agent to reach the goal, resulting in a more heavily discounted reward. Conversely, as the entropy of actions at

each state is increased, there is an increase in the mutual information between the actions and the compressed message during the minimum entropy coupling at each step. This dynamic establishes a fundamental trade-off between the MDP reward and the receiver's decoding accuracy, through the adjustment of $\beta$. Figure 8 shows policies learned by high and low values of $\beta$.

We compare our proposed compression method in Algorithm 1 with a baseline of uniform quantization. As detailed in Algorithm 5, given an entropy budget $R$, the input symbols are uniformly partitioned into $\lfloor 2^R \rfloor$ bins, and each bin is encoded with the same code.

Figure 4 illustrates the trade-off between the average MDP reward and the receiver's decoding accuracy by varying $\beta$, using our deterministic EBIM solver in Algorithm 1, and the uniform quantization encoder in Algorithm 5. Here, the compression rate is defined by the ratio of the message entropy to the allowed code budget $H(T)$.

---

**Algorithm 5** Uniform Quantizer Encoder

**Input:** $p_X, R$
**Output:** $p_{XT}$
1:   $n \leftarrow$ length of $p_X$
2:   $m \leftarrow \lfloor 2^R \rfloor$
3:   partition_size $\leftarrow \lceil n/m \rceil$
4:   Initialize $p_{XT}$ as an $n \times m$ zero matrix
5:   **for** $i \leftarrow 0$ **to** $m-1$ **do**
6:      $start \leftarrow i \times$ partition_size
7:      $end \leftarrow \min(start + \text{partition\_size}, n)$
8:      $p_{XT}[start:end, i] \leftarrow p_X[start:end]$
9:   **return** $p_{XT}$

---

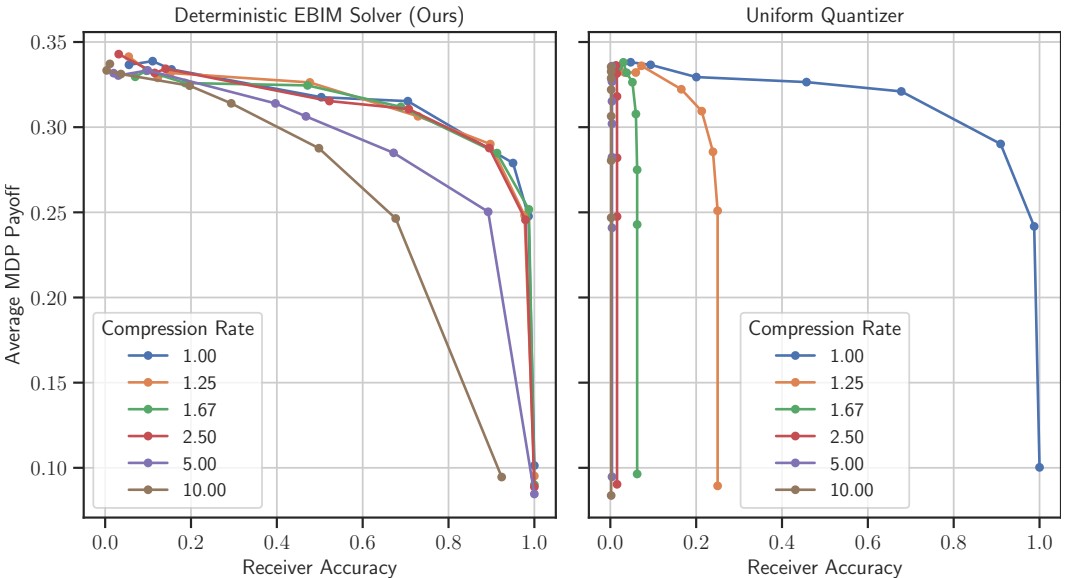

Figure 4: The trade-off between average MDP reward vs. receiver's accuracy, navigated by varying the value of $\beta$. Left: using our search algorithm for compression (Algorithm 1), Right: using uniform quantization in Algorithm 5. The message size is 512 with a uniform prior, and each data point is averaged over 200 episodes.

Figure 5 illustrates the evolution of message belief over time for various values of $\beta$ and rate budgets. A marginal policy optimized with a higher $\beta$ prioritizes message accuracy over MDP payoff, as higher entropy of actions at each state provides more room for the agent to encode information about the message. Consequently, as observed, this leads to improved receiver accuracy in fewer steps. In addition, a lower compression rate permits the agent to retain more information about the message, enabling more effective encoding of information in the selected trajectory.

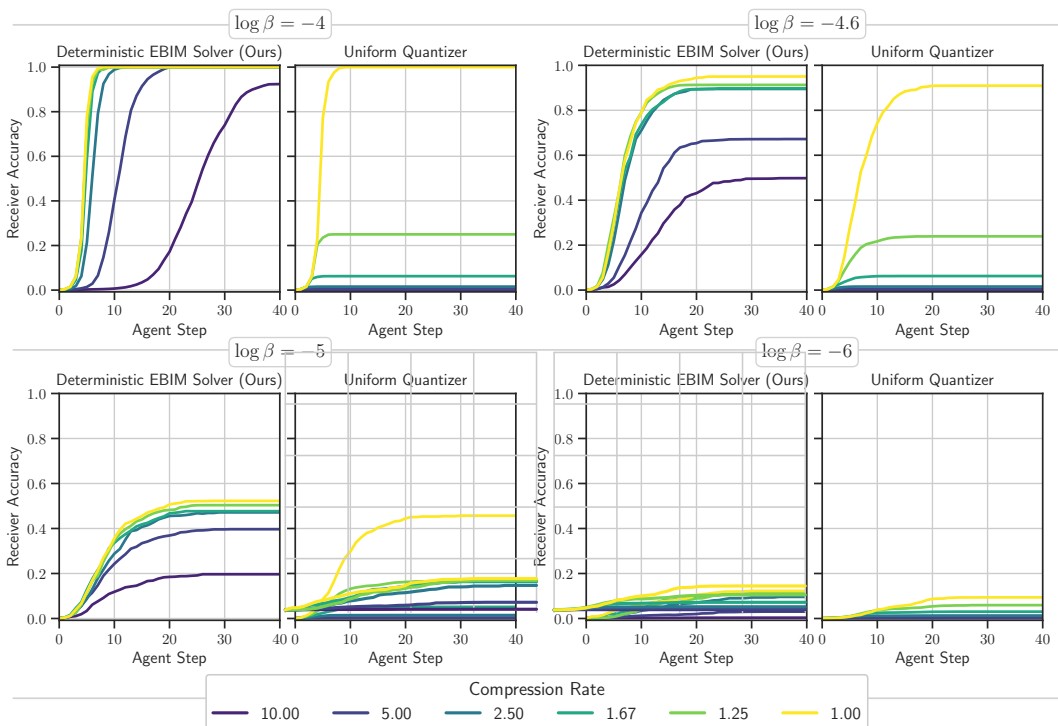

Figure 5: Evolution of message belief over time, for various values of $\beta$ and rate budget, using our search algorithm for compression in Algorithm 1 vs. uniform quantization in Algorithm 5.

## 5 Conclusion

We investigated a lossy compression framework under logarithmic loss, where the reconstruction distribution differs from the source distribution. This framework supports joint compression and retrieval applications, or more generally, cases where distributional shifts occur due to processing. We demonstrated that this framework effectively extends the classical minimum entropy coupling by incorporating a bottleneck, which regulates the degree of stochasticity in the coupling.

Furthermore, we showed that separately optimizing the encoder and decoder decomposes the Minimum Entropy Coupling with Bottleneck (MEC-B) into two distinct problems: Entropy-Bounded Information Maximization (EBIM) for the encoder, followed by Minimum Entropy Coupling (MEC) for the decoder. We conducted an extensive study of the EBIM problem, provided a functional mapping search algorithm with guaranteed performance, and characterized the optimal solution adjacent to functional mappings, offering valuable theoretical insights into the problem structure. To illustrate an application of MEC-B, we presented experiments on Markov Coding Games (MCGs) with rate limits. The results demonstrated the trade-off between MDP reward and receiver accuracy, with varying compression rates, compared to baseline compression schemes.

Future research could focus on quantifying the gap between the separate optimization of the encoder and decoder and the optimal joint setting. Also, enabling fine-grained control over the entropy spread in the coupling can be key in some applications. Additionally, the application of Entropy-Bounded Information Maximization (EBIM) in watermarking language models [29] suggests a valuable intersection with state-of-the-art AI applications. Moreover, extending this framework to continuous cases could lead to the design of neural network architectures based on the proposed framework and provide information-theoretic insights into a broad spectrum of deep learning problems. These include unpaired sample-to-sample translation [30–32], joint compression and upscaling [25, 33], and the InfoMax framework [34, 35], among others.

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

# A   Mathematical Proofs

## A.1   Proof of Lemma 1

**Lemma 1.** *Given a Markov chain $X \leftrightarrow T \leftrightarrow Y$:*

$$I(X;Y) = I(X;T) + I(Y;T) - I(T;X,Y). \tag{5}$$

*Proof.* By multiple applications of the chain rule for mutual information, i.e. $I(A;B,C) = I(A;B) + I(A;C|B)$, we have:

$$I(X;Y) = I(X;Y,T) - I(X;T|Y) \tag{15}$$
$$= [I(X;T) + I(X;Y|T)] - [-I(T;Y) + I(T;X,Y)] \tag{16}$$
$$= I(X;T) + I(Y;T) - I(T;X,Y). \tag{17}$$

Note that from the Markov chain property, we have $I(X;Y|T) = 0$.    □

## A.2   Proof of Theorem 1

**Theorem 1.** $\mathcal{I}_{EBIM}(p_X, R) = R$ *if and only if there exists a function* $g : \mathcal{X} \to \mathcal{T}$ *such that* $H(g(X)) = R$.

*Proof.* If such $g$ exists, let

$$p^*_{XT}(x,t) = \begin{cases} p_X(x) & t = g(x) \\ 0 & \text{otherwise} \end{cases}$$

This joint distribution effectively sets $T = g(X)$. Note that $p^*_{XT} \in \mathcal{M}$ and we have $I(X;T) = H(T) - H(T|X) = H(g(X)) = R$. Since $\mathcal{I}_{\text{EBIM}}(p_X, R) \leq R$, we conclude that $\mathcal{I}_{\text{EBIM}}(p_X, R) = R$ for $p_{XT} = p^*_{XT}$.

Conversely, if $\mathcal{I}_{\text{EBIM}}(p_X, R) = R$, then there exists $p^*_{XT} \in \mathcal{M}$ such that $I(X;T) = R$. Therefore

$$H(T) = I(X;T) + H(T|X) = R + H(T|X) \leq R$$
$$\Rightarrow H(T|X) \leq 0$$

As a result, $H(T|X) = 0$ and $H(T) = R$, which means $p^*_{XT}$ defines a function $g$ such that $T = g(X)$, and $H(g(X)) = H(T) = R$.    □

## A.3   Proof of Theorem 2

Before providing the formal proof, it is helpful to gain some insight into the structure of the solution first.

**Remark 2.** *Let $n = |p_X|$. The procedure outlined in Algorithm 1 establishes a series of deterministic mappings $p_l^{(0)}, p_s^{(1)}, p_l^{(1)}, \cdots, p_s^{(n-1)}, p_l^{(n-1)}$, corresponding to a decreasing sequence of mutual information values $I_l^{(0)}, I_s^{(1)}, I_l^{(1)}, \cdots, I_s^{(n-1)}, I_l^{(n-1)}$. The algorithm then picks the mapping with the maximum mutual information that does not exceed R. Therefore*

$$R - I(X;T) \leq \max\left\{I_l^{(0)} - I_s^{(1)}, \ I_s^{(1)} - I_l^{(1)}, \ \cdots, \ I_s^{(n-1)} - I_l^{(n-1)}\right\}$$
$$\leq \max\left\{I_l^{(0)} - I_l^{(1)}, \ I_l^{(1)} - I_l^{(2)}, \ \cdots, \ I_l^{(n-2)} - I_l^{(n-1)}\right\}. \tag{18}$$

**Example.** *For $p_X = [0.4\ \ 0.3\ \ 0.2\ \ 0.1]$, Algorithm 1 traverses through the following deterministic mappings, from left to right:*

| | $p_l^{(0)}$ | $p_s^{(1)}$ | $p_l^{(1)}$ | $p_s^{(2)}$ | $p_l^{(2)}$ | $p_s^{(3)}$ | $p_l^{(3)}$ |
|---|---|---|---|---|---|---|---|
| $p_{XT}$ | $\begin{bmatrix} 0.4 & 0 & 0 & 0 \\ 0 & 0.3 & 0 & 0 \\ 0 & 0 & 0.2 & 0 \\ 0 & 0 & 0 & 0.1 \end{bmatrix}$ | $\begin{bmatrix} 0.4 & 0 & 0 \\ 0 & 0.3 & 0 \\ 0 & 0 & 0.2 \\ 0 & 0 & 0.1 \end{bmatrix}$ | $\begin{bmatrix} 0.4 & 0 & 0 \\ 0.3 & 0 & 0 \\ 0 & 0.2 & 0 \\ 0 & 0 & 0.1 \end{bmatrix}$ | $\begin{bmatrix} 0.4 & 0 \\ 0.3 & 0 \\ 0 & 0.2 \\ 0 & 0.1 \end{bmatrix}$ | $\begin{bmatrix} 0.4 & 0 \\ 0.3 & 0 \\ 0.2 & 0 \\ 0 & 0.1 \end{bmatrix}$ | $\begin{bmatrix} 0.4 \\ 0.3 \\ 0.2 \\ 0.1 \end{bmatrix}$ | $\begin{bmatrix} 0.4 \\ 0.3 \\ 0.2 \\ 0.1 \end{bmatrix}$ |
| $I(X;T)$ | $H(p_X)$ | $H([.4\ \ .3\ \ .3])$ | $H([.7\ \ .2\ \ .1])$ | $H([.7\ \ .3])$ | $H([.9\ \ .1])$ | $0$ | $0$ |

**Definition 1.** *Let $P'$ be a probability distribution resulted from merging two elements $p > 0$ and $q > 0$ in an original distribution $P$, i.e. $P = [\cdots \ \ p \ \ \cdots \ \ q \ \ \cdots]$ and $P' = [\cdots \ \ p + q \ \ \cdots]$. Then, the amount of decrease in the entropy from this merge operation is characterized by:*

$$\Delta_H(p, \, q) = H(P) - H(P') \tag{19}$$

$$= p \log \frac{1}{p} + q \log \frac{1}{q} - (p + q) \log \frac{1}{p + q} \tag{20}$$

$$= p \log \left(1 + \frac{q}{p}\right) + q \log \left(1 + \frac{p}{q}\right). \tag{21}$$

**Lemma 2.** *The following properties hold for the function $\Delta_H$:*

1. *$\Delta_H(\cdot, \, \cdot)$ is monotonically increasing in both arguments.*

2. *$\Delta_H(\cdot, \, \cdot)$ is concave.*

3. *$\Delta_H(p, \, 1 - p) = h(p)$.*

*Proof.* The properties are derived through straightforward derivative calculations:

1. $\frac{\partial}{\partial p}\Delta_H = \log(1 + q/p) \geq 0$, and $\frac{\partial}{\partial q}\Delta_H = \log(1 + p/q) \geq 0$.

2. The Hessian of $\Delta_H$ is negative semidefinite:

$$\mathbf{H}_{\Delta_H} = \frac{1}{p + q} \begin{bmatrix} -q/p & 1 \\ 1 & -p/q \end{bmatrix}, \tag{22}$$

   with eigenvalues $\lambda_1 = 0$ and $\lambda_2 = -(\frac{q}{p} + \frac{p}{q})(\frac{1}{p+q}) < 0$.

3. $\Delta_H(p, \, 1 - p) = -p \log p - (1 - p) \log(1 - p) = h(p)$.

$\square$

**Theorem 2.** *If the output of Algorithm 1 yields mutual information $\widehat{I}$, then*

$$\mathcal{I}_{EBIM}(p_X, R) - \widehat{I} \leq h(p_2), \tag{13}$$

*where $h(\cdot)$ is the binary entropy function, and $p_2$ denotes the second largest element of $p_X$.*

*Proof.* For the gap to the optimal objective, $\mathcal{I}_{\text{EBIM}}(p_X, R) - \widehat{I}$, we have:

$\triangleright$ Equation (11) $\qquad \mathcal{I}_{\text{EBIM}}(p_X, R) - \widehat{I} \leq R - \widehat{I}$

$\triangleright$ Remark 2 $\qquad\qquad\qquad\quad \leq \max\limits_{i \in \{1, \cdots, n-1\}} I_l^{(i-1)} - I_l^{(i)}$

$\triangleright$ Algorithm 1, Line 8 $\qquad = \max\limits_{i \in \{1, \cdots, n-1\}} H\left(\sum\limits_x p_l^{(i-1)}\right) - H\left(\sum\limits_x p_l^{(i)}\right)$

$\triangleright$ Definition 1 $\qquad\qquad\quad = \max\limits_{i \in \{1, \cdots, n-1\}} \Delta_H\left(\sum\limits_{k=1}^{i} p_k\,,\ p_{i+1}\right)$

$\triangleright$ Lemma 2.1 $\qquad\qquad\quad \leq \max\limits_{i \in \{1, \cdots, n-1\}} \Delta_H\left(\sum\limits_{k=1}^{i} p_k + \sum\limits_{k=i+2}^{n} p_k\,,\ p_{i+1}\right)$

$\qquad\qquad\qquad\qquad\qquad = \max\limits_{i \in \{1, \cdots, n-1\}} \Delta_H\left(1 - p_{i+1}\,,\ p_{i+1}\right)$

$\triangleright$ Lemma 2.3 $\qquad\qquad\quad = \max\limits_{i \in \{1, \cdots, n-1\}} h\left(p_{i+1}\right)$

$\triangleright$ $p_2, p_3, \cdots, p_n \leq 0.5 \qquad = h(p_2).$

$\qquad\qquad\qquad\qquad\qquad\qquad\qquad\qquad\qquad\qquad\qquad\qquad\qquad\qquad \square$

Note that the above bound on the optimality of the proposed algorithm is by no means tight, as it does not account for the intermediate distributions $p_s^{(i)}$.

### A.4 Proof of Theorem 3

**Theorem 3.** *Let $p_{XT}$ denoted by a $|\mathcal{X}| \times |\mathcal{T}|$ matrix, defines a deterministic mapping $T = g(X)$, with $I(X; T) = H(T) = R_g$. We have $\mathcal{I}_{EBIM}(p_X, R_g) = R_g$, and for small enough $\epsilon > 0$:*

1. *$\mathcal{I}_{EBIM}(p_X, R_g + \epsilon)$ is attained as follows:*
   *Normalize the columns by dividing each column by its sum. Then, select the cell with the smallest normalized value and move an infinitesimal probability mass from this cell to a new column of $p_{XT}$ in the same row.*

2. *$\mathcal{I}_{EBIM}(p_X, R_g - \epsilon)$ is achieved as follows:*
   *Identify the columns with the smallest and largest sums in $p_{XT}$. Select the cell with the smallest value in the column with the lowest sum. Transfer an infinitesimal probability mass from this cell to the column with the highest sum in the same row.*

**Example.** *Figure 6 depicts an example of optimal solutions in the neighborhood of a deterministic mapping.*

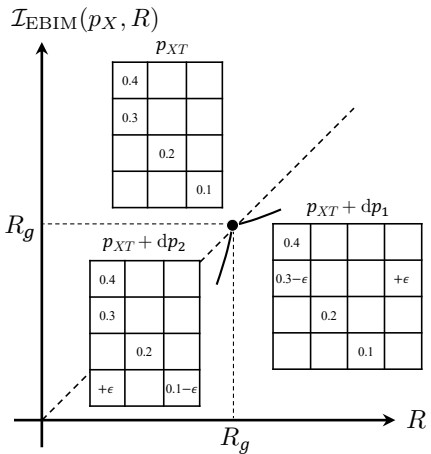

Figure 6: Optimal solutions in the neighborhood of a deterministic mapping.

*Proof.* Let us view the joint distribution as a $n \times m$ matrix $p_{XT}$. Note that:

$$p_{XT}(x,t) = \begin{cases} p_X(x) & t = g(x) \\ 0 & \text{otherwise} \end{cases} \tag{23}$$

For example:

$$p_{XT} = \begin{bmatrix} 0.4 & 0 & 0 & 0 \\ 0.3 & 0 & 0 & 0 \\ 0 & 0.2 & 0 & 0 \\ 0 & 0 & 0.1 & 0 \end{bmatrix}, \quad g(x) = \begin{cases} 1, & x = 1,2 \\ 2, & x = 3 \\ 3, & x = 4 \end{cases} \tag{24}$$

Consider a perturbation $\mathrm{d}P \in \mathbb{R}^{n \times m}$ to $p_{XT}$. For $p_{XT} + \mathrm{d}P$ to be a valid distribution in $\mathcal{M}$, we need:

1. $\sum_t \mathrm{d}P(x,t) = 0, \quad \forall x \in \mathcal{X}$

2. $\mathrm{d}P(x,t) \geq 0, \quad \forall x,t \text{ s.t. } t \neq g(x)$

3. $\mathrm{d}P(x,t) \leq 0, \quad \forall x,t \text{ s.t. } t = g(x)$

We define the set of all such perturbations as $\Omega \subset \mathbb{R}^{n \times m}$. Next, let us define basis perturbations $\Delta_{x,t}$ for $t \neq g(x)$ as:

$$[\Delta_{x,t}]_{ij} = \begin{cases} -\varepsilon, & \text{if } i = x, \; j = g(x) \\ +\varepsilon, & \text{if } i = x, \; j = t \\ 0, & \text{otherwise} \end{cases} \tag{25}$$

Note that $\Delta_{x,t}$ represents moving a probability mass of $\varepsilon$ from non-zero cell $(x, g(x))$ to empty cell $(x,t)$. For the example in equation (24):

$$\Delta_{2,3} = \begin{bmatrix} 0 & 0 & 0 & 0 \\ -\varepsilon & 0 & +\varepsilon & 0 \\ 0 & 0 & 0 & 0 \\ 0 & 0 & 0 & 0 \end{bmatrix}$$

The significance of these bases is that any perturbation in $\Omega$ can be represented as:

$$\mathrm{d}P = \sum_{\substack{x,t \\ t \neq g(x)}} \alpha_{x,t} \, \Delta_{x,t}, \tag{26}$$

with coefficients $\alpha_{x,t} \geq 0$. For example:

$$\begin{bmatrix} 0 & 0 & 0 & 0 \\ -3\varepsilon & +2\varepsilon & +\varepsilon & 0 \\ 0 & 0 & 0 & 0 \\ 0 & 0 & 0 & 0 \end{bmatrix} = 2 \times \Delta_{1,1} + \Delta_{1,2}.$$

Realizing $I_{XT}$, $H_{XT}$, and $H_T$ as functions of a joint distribution, we are interested in calculating the ratio $\mathrm{d}I_{XT}/\mathrm{d}H_T$ with respect to a perturbation $\mathrm{d}P \in \Omega$ as $\varepsilon \to 0$ at $p_{XT}$. Note that since for any $\mathrm{d}P \in \Omega$, $\mathrm{d}H_X = 0$, we have:

$$\frac{\mathrm{d}I_{X,T}}{\mathrm{d}H_T} = \frac{\mathrm{d}H_X + \mathrm{d}H_T - \mathrm{d}H_{X,T}}{\mathrm{d}H_T} = 1 - \frac{\mathrm{d}H_{X,T}}{\mathrm{d}H_T}.$$

Therefore

$$\frac{\mathrm{d}I_{X,T}}{\mathrm{d}H_T} = 1 - \frac{\mathrm{d}H_{X,T}\left(p_{XT}, \mathrm{d}P\right)}{\mathrm{d}H_T\left(p_{XT}, \mathrm{d}P\right)}$$

$$= 1 - \frac{\mathrm{d}H_{X,T}\left(p_{XT}, \sum_x \sum_{t \neq g(x)} \alpha_{x,t}\, \Delta_{x,t}\right)}{\mathrm{d}H_T\left(p_{XT}, \sum_x \sum_{t \neq g(x)} \alpha_{x,t}\, \Delta_{x,t}\right)}$$

$$= 1 - \frac{\sum_x \sum_{t \neq g(x)} \alpha_{x,t}\, \mathrm{d}H_{X,T}\left(p_{XT}, \Delta_{x,t}\right)}{\sum_x \sum_{t \neq g(x)} \alpha_{x,t}\, \mathrm{d}H_T\left(p_{XT}, \Delta_{x,t}\right)}. \tag{27}$$

$\mathrm{d}H_{X,T}\left(p_{XT}, \Delta_{x,t}\right)$ represents the amount of change in the joint entropy, when an infinitesimal mass of $\varepsilon$ is moved from $(x, g(x))$ to $(x, t)$. More precisely, from (23) and (25):

$$\mathrm{d}H_{X,T}\left(p_{XT}, \Delta_{x,t}\right) = H_{X,T}\left(p_{XT} + \Delta_{x,t}\right) - H_{X,T}\left(p_{XT}\right)$$

$$= \left[-(p_X(x) - \varepsilon)\log(p_X(x) - \varepsilon) - \varepsilon\log\varepsilon\right] - \left[-p_X(x)\log p_X(x)\right]$$

$$= p_X(x)\log\frac{p_X(x)}{p_X(x) - \varepsilon} + \varepsilon\log\frac{p_X(x) - \varepsilon}{\varepsilon}$$

$$= \varepsilon + \mathcal{O}(\varepsilon^2) + \varepsilon\log\frac{p_X(x)}{\varepsilon}. \tag{28}$$

The last line uses the fact that for small enough $x$, $f(x) = a\log\frac{a}{a-x} = x + \mathcal{O}(x^2)$. Similarly:

$$\mathrm{d}H_T\left(p_{XT}, \Delta_{x,t}\right) = H_T\left(p_{XT} + \Delta_{x,t}\right) - H_T\left(p_{XT}\right)$$

$$= \left[-\left(p_T\big(g(x)\big) - \varepsilon\right)\log\left(p_T\big(g(x)\big) - \varepsilon\right) - \left(p_T(t) + \varepsilon\right)\log\left(p_T(t) + \varepsilon\right)\right]$$

$$- \left[-p_T\big(g(x)\big)\log p_T\big(g(x)\big) - p_T(t)\log p_T(t)\right]$$

$$= p_T\big(g(x)\big)\log\frac{p_T\big(g(x)\big)}{p_T\big(g(x)\big) - \varepsilon} + p_T(t)\log\frac{p_T(t)}{p_T(t) + \varepsilon} + \varepsilon\log\frac{p_T\big(g(x)\big) - \varepsilon}{p_T(t) + \varepsilon}$$

$$= \varepsilon + \mathcal{O}(\varepsilon^2) - \varepsilon + \mathcal{O}(\varepsilon^2) + \varepsilon\log\frac{p_T\big(g(x)\big)}{p_T(t) + \varepsilon}$$

$$= \log\frac{p_T\big(g(x)\big)}{p_T(t) + \varepsilon} + \mathcal{O}(\varepsilon^2). \tag{29}$$

Plugging (28) and (29) back to (27), we will get:

$$\frac{\mathrm{d}I_{X,T}}{\mathrm{d}H_T} = 1 - \frac{\sum_x \sum_{t \neq g(x)} \alpha_{x,t}\left[\varepsilon + \varepsilon\log\frac{p_X(x)}{\varepsilon} + \mathcal{O}(\varepsilon^2)\right]}{\sum_x \sum_{t \neq g(x)} \alpha_{x,t}\left[\log\frac{p_T\big(g(x)\big)}{p_T(t) + \varepsilon} + \mathcal{O}(\varepsilon^2)\right]}$$

$$= 1 - \frac{\sum_x \sum_{t \neq g(x)} \overline{\alpha}_{x,t}\log\frac{p_X(x)}{\varepsilon}}{\sum_x \sum_{t \neq g(x)} \overline{\alpha}_{x,t}\log\frac{p_T\big(g(x)\big)}{p_T(t) + \varepsilon}}, \tag{30}$$

where $\alpha_{x,t}$ is normalized by:

$$\overline{\alpha}_{x,t} = \frac{\alpha_{x,t}}{\sum\limits_{x}\sum\limits_{t \neq g(x)} \alpha_{x,t}}.$$

Let's focus on the limit of (30) when $\varepsilon \to 0$: If there is any $t \in \mathcal{T}$ with $p_T(t) = 0$ and $\overline{\alpha}_{x,t} > 0$, the denominator of the second term will grow without bound, otherwise the denominator remains bounded. Therefore, for the limit of (30) we have:

$$\lim_{\varepsilon \to 0} \frac{\mathrm{d}I_{X,T}}{\mathrm{d}H_T} = \begin{cases} -\infty & \text{if} \quad \overline{\alpha}_{x,t} = 0 \quad \forall t \text{ s.t. } p_T(t) = 0 \\ 1 - \Big(\sum\limits_{x}\sum\limits_{p_T(t)=0} \overline{\alpha}_{x,t}\Big)^{-1} & \text{if} \quad \exists t : \overline{\alpha}_{x,t} > 0 \text{ and } p_T(t) = 0 \end{cases} \tag{31}$$

For $\mathrm{d}H_T > 0$, we need to find a perturbation (i.e. coefficients $\alpha_{x,t}$) that maximizes $\mathrm{d}I_{XT}/\mathrm{d}H_T$. From (31), this means $\exists t \in \mathcal{T}$ with $\overline{\alpha}_{x,t} > 0$ and $p_T(t) = 0$.

$$\overline{\alpha} = \operatorname*{argmax}_{\overline{\alpha}} \frac{\mathrm{d}I_{X,T}}{\mathrm{d}H_T} = \operatorname*{argmax}_{\overline{\alpha}} 1 - \Big(\sum_{x}\sum_{p_T(t)=0} \overline{\alpha}_{x,t}\Big)^{-1}$$

$$= \operatorname*{argmax}_{\overline{\alpha}} \sum_{x}\sum_{p_T(t)=0} \overline{\alpha}_{x,t}$$

Therefore, $\sum\limits_{x}\sum\limits_{p_T(t)=0} \overline{\alpha}_{x,t} = 1$ which means $\overline{\alpha}_{x,t} = 0$ if $p_T(t) > 0$. In other words, we should only consider perturbations where masses are moved to all-zero columns. Continuing (30):

$$\overline{\alpha} = \operatorname*{argmax}_{\overline{\alpha}} \frac{\mathrm{d}I_{X,T}}{\mathrm{d}H_T}$$

$$= \operatorname*{argmax}_{\overline{\alpha}} 1 - \frac{\sum\limits_{x}\sum\limits_{p_T(t)=0} \overline{\alpha}_{x,t} \log \dfrac{p_X(x)}{\varepsilon}}{\sum\limits_{x}\sum\limits_{p_T(t)=0} \overline{\alpha}_{x,t} \log \dfrac{p_T\big(g(x)\big)}{\varepsilon}}$$

$$= \operatorname*{argmax}_{\overline{\alpha}} 1 - \frac{-\log \varepsilon + \sum\limits_{x}\sum\limits_{p_T(t)=0} \overline{\alpha}_{x,t} \log p_X(x)}{-\log \varepsilon + \sum\limits_{x}\sum\limits_{p_T(t)=0} \overline{\alpha}_{x,t} \log p_T\big(g(x)\big)}$$

$$= \operatorname*{argmax}_{\overline{\alpha}} \frac{-1}{\log \varepsilon} \left[\sum_{x}\sum_{p_T(t)=0} \overline{\alpha}_{x,t} \log \frac{p_T\big(g(x)\big)}{p_X(x)}\right]$$

$$= \operatorname*{argmin}_{\overline{\alpha}} \sum_{x}\sum_{p_T(t)=0} \overline{\alpha}_{x,t} \log \frac{p_X(x)}{p_T\big(g(x)\big)}.$$

This is achieved by selecting

$$\Rightarrow \alpha_{x,t} = \begin{cases} 1, & x = \operatorname*{argmin}_{x'} \dfrac{p_X(x')}{p_T\big(g(x')\big)} \text{ and } p_T(t) = 0 \\ 0, & \text{otherwise} \end{cases} \tag{32}$$

In other words, first, normalize columns in $p_{XT}$ by their sum, then move an infinitesimal probability mass from the cell with the smallest normalized value to an all-zero column. It is easy to confirm that $\mathrm{d}H_T > 0$ for such a perturbation. For the example distribution in (24):

$$p_{XT} + \mathrm{d}P = \begin{bmatrix} 0.4 & 0 & 0 & 0 \\ 0.3 - \varepsilon & 0 & 0 & \varepsilon \\ 0 & 0.2 & 0 & 0 \\ 0 & 0 & 0.1 & 0 \end{bmatrix} \tag{33}$$

On the other hand, for $\mathrm{d}H_T < 0$, we need to find a perturbation (i.e. coefficients $\alpha_{x,t}$) that minimizes $\mathrm{d}I_{XT}/\mathrm{d}H_T$. From (31), this means $\overline{\alpha}_{x,t} = 0$ for all $t \in \mathcal{T}$ that $p_T(t) = 0$. Therefore, as in (30):

$$
\begin{aligned}
\overline{\alpha} &= \operatorname*{argmin}_{\overline{\alpha}} \; \frac{\mathrm{d}I_{X,T}}{\mathrm{d}H_T} \\[2em]
&= \operatorname*{argmin}_{\overline{\alpha}} \; 1 - \frac{\displaystyle\sum_x \sum_{t \neq g(x)} \overline{\alpha}_{x,t} \log \frac{p_X(x)}{\varepsilon}}{\displaystyle\sum_x \sum_{t \neq g(x)} \overline{\alpha}_{x,t} \log \frac{p_T\big(g(x)\big)}{p_T(t)}} \\[2em]
&= \operatorname*{argmin}_{\overline{\alpha}} \; 1 - \frac{-\log \varepsilon + \displaystyle\sum_x \sum_{t \neq g(x)} \overline{\alpha}_{x,t} \log p_X(x)}{\displaystyle\sum_x \sum_{t \neq g(x)} \overline{\alpha}_{x,t} \log \frac{p_T\big(g(x)\big)}{p_T(t)}} \\[2em]
&= \operatorname*{argmin}_{\overline{\alpha}} \; \sum_x \sum_{t \neq g(x)} \overline{\alpha}_{x,t} \log \frac{p_T\big(g(x)\big)}{p_T(t)}.
\end{aligned}
$$

This is achieved by selecting

$$
\Rightarrow \overline{\alpha}_{x,t} = \begin{cases} 1, & x = \operatorname*{argmin}_{x'} p_T\big(g(x')\big) \ \text{ and } \ t = \operatorname*{argmax}_{t'} p_T(t') \\ 0, & \text{otherwise} \end{cases} \tag{34}
$$

In other words, moving an infinitesimal probability mass from the smallest column to the largest column of $p_{XT}$. It is easy to confirm that $\mathrm{d}H_T < 0$ for such a perturbation. For the example distribution of (24):

$$
p_{XT} + \mathrm{d}P = \begin{bmatrix} 0.4 & 0 & 0 & 0 \\ 0.3 & 0 & 0 & 0 \\ 0 & 0.2 & 0 & 0 \\ \varepsilon & 0 & 0.1 - \varepsilon & 0 \end{bmatrix} \tag{35}
$$

$\square$

# B   Minimum Entropy Coupling

Consider two discrete random variables $X$ and $Y$, over alphabets $\mathcal{X}$ and $\mathcal{Y}$ with probability mass functions $p_X$ and $p_Y$, respectively. The goal of minimum entropy coupling is to find the joint distribution $p_{XY}$ that minimizes the joint entropy $H(X,Y)$:

$$
\begin{aligned}
\min_{p_{XY}} \; & H(X;Y) \\
\text{s.t.} \; & \sum_{y \in \mathcal{Y}} p_{XY}(x,y) = p_X(x) \quad \forall x \in \mathcal{X}, \\
& \sum_{x \in \mathcal{X}} p_{XY}(x,y) = p_Y(y) \quad \forall y \in \mathcal{Y}
\end{aligned} \tag{36}
$$

This is a concave minimization problem over a standard polyhedron [36]. Therefore, every vertex of the polyhedron is a local minimum and the global minimum happens at a subset of the vertices.

Note that an standard polyhedron is defined as $\mathcal{P} = \{\boldsymbol{x} \in \mathbb{R}^n \,|\, \boldsymbol{A}\boldsymbol{x} = \boldsymbol{b}, \ \boldsymbol{x} \geq \boldsymbol{0}\}$, where $\boldsymbol{A} \in \mathbb{R}^{m \times n}$ with linearly independent rows. A point $\boldsymbol{x}^* \in \mathcal{P}$ is a vertex if and only if it has $n - m$ zero elements and columns of $\boldsymbol{A}$ corresponding to other $m$ non-zero elements are linearly independent. Hence, to exhaustively iterate all the vertices:

1. Choose $m$ linearly independent columns $\boldsymbol{A}_{\pi(1)}, \cdots, \boldsymbol{A}_{\pi(m)}$.

2. Let $\boldsymbol{x}_i = 0$ for all $i \in \pi(1), ..., \pi(m)$

3. Solve the system of $m$ equations $\boldsymbol{Ax} - \boldsymbol{b} = 0$ for the unknowns $\boldsymbol{x}_\pi(1), \cdots, \boldsymbol{x}_\pi(m)$

Therefore a crude upper-bound on the number of vertices would be $\binom{n}{m}$. This can be enhanced to $\binom{n-m/2}{m/2}$ [37] which is still exponential in $m$. Next, we will show that the minimum entropy coupling problem as defined in (36) is essentially NP-Hard. This is done by reduction from another NP-complete problem, the k-Subset-Sum (see [6] for more details).

**Remark 3.** *The minimum entropy coupling problem in (36) is NP-Hard [6].*

*Proof.* To show an optimization problem is NP-Hard, we need to show the corresponding decision problem is NP-Hard. Given an optimization problem, a decision version is whether or not any target value $t$ is achievable. Without the loss of generality, assume $|\mathcal{X}| > |\mathcal{Y}|$. We set $t = H(Y)$, i.e. to decide if there exists a function $f : \mathcal{X} \to \mathcal{Y}$ such that $Y = f(X)$. Let's call this problem *Deterministic Matching*.

Next, we show any instance of the $k$-Subset-Sum problem can be reduced to an instance of Deterministic Matching, by a polynomial-time procedure (denoted by the notation $<_p$). Consider a general instance of the $k$-Subset-Sum problem: Given set $\mathcal{S}$ of integers and target values $\{t_i | 1 \leq i \leq k\}$, decide if there exists a partition $\{\mathcal{S}_i | 1 \leq i \leq k\}$ of size $k$ on $\mathcal{S}$ such that $\sum \mathcal{S}_i = t_i$ for all $1 \leq i \leq k$. Now, set $p_X(i) = s_i / \sum(\mathcal{S}), \forall s_i \in \mathcal{S}$ and $p_Y(i) = t_i / \sum(t_j)$. Then, clearly solving Deterministic Matching for $p_X, p_Y$ will solve the original $k$-Subset-Sum problem. Therefore, $k$-Subset-Sum $<_p$ Deterministic Matching and hence, Deterministic Matching is NP-Hard. Consequently, Minimum Entropy Coupling is an NP-Hard optimization problem. □

Finally, we introduce two linear-time approximate greedy algorithms for the minimum entropy coupling problem, and numerically compare their achieved minima to a general approximate algorithm.

---

**Algorithm 6** Max-Seeking Minimum Entropy Coupling

---

**Input:** marginal distributions $p_X, p_Y$
**Output:** joint distribution $p_{XY}$
1: $p_{XY}(x, y) \leftarrow 0, \quad \forall x, y \in \mathcal{X}, \mathcal{Y}$
2: **while** $p_X, p_Y \neq \boldsymbol{0}$ **do**
3: $\quad x^* \leftarrow \text{argmax}_x \, p_X(x)$
4: $\quad y^* \leftarrow \text{argmax}_y \, p_Y(y)$
5: $\quad p_{XY}(x^*, y^*) \leftarrow \min\{p_X(x^*), p_Y(y^*)\}$
6: $\quad p_X(x^*) \leftarrow p_X(x^*) - \min\{p_X(x^*), p_Y(y^*)\}$
7: $\quad p_Y(y^*) \leftarrow p_Y(y^*) - \min\{p_X(x^*), p_Y(y^*)\}$
8: **return** $p_{XY}$

---

---

**Algorithm 7** Zero-Seeking Minimum Entropy Coupling

---

**Input:** marginal distributions $p_X, p_Y$
**Output:** joint distribution $p_{XY}$
1: $p_{XY}(x, y) \leftarrow 0, \quad \forall x, y \in \mathcal{X}, \mathcal{Y}$
2: **while** $p_X, p_Y \neq \boldsymbol{0}$ **do**
3: $\quad (x^*, y^*) \leftarrow \text{argmin}_{x,y} \, |p_X(x) - p_Y(y)|$
4: $\quad p_{XY}(x^*, y^*) \leftarrow \min\{p_X(x^*), p_Y(y^*)\}$
5: $\quad p_X(x^*) \leftarrow p_X(x^*) - \min\{p_X(x^*), p_Y(y^*)\}$
6: $\quad p_Y(y^*) \leftarrow p_Y(y^*) - \min\{p_X(x^*), p_Y(y^*)\}$
7: **return** $p_{XY}$

---

At each step, each algorithm selects a symbol from each random variable and connects them in the joint distribution by assigning the higher probability of the two symbols, updating the marginals accordingly. The max-seeking version targets the symbols with the largest remaining probability mass

at each step, whereas the zero-seeking version pairs symbols with the most similar probability mass. Furthermore, the greedy algorithm described by Kocaoglu et al. [14] resembles the max-seeking version outlined in Algorithm 6.

As a simple baseline, we randomly generated 100 pairs of joint distributions and fed them to our greedy solvers. We also used a general concave minimization method, Successive Linearization Algorithm (SLA) [38], and compared the achieved joint entropy. Table 1 summarizes the average joint entropy over 100 trials for each method.

Table 1: Minimum Entropy Coupling: average achieved joint entropy of 100 simulations of marginal distributions.

| Name | Entropy |
|------|---------|
| Independent Joint | $5.443 \pm 0.101$ |
| SLA | $3.225 \pm 0.141$ |
| Max-Seeking Greedy | $2.946 \pm 0.064$ |
| Zero-Seeking Greedy | $2.937 \pm 0.058$ |

## C   Markov Coding Games

### C.1   Backgrounds and Notations

**Markov Decision Process** MDPs are represented by the tuple notation $(\mathcal{S}, \mathcal{A}, \mathcal{R}, \mathcal{T})$, where $\mathcal{S}$ is the state space, $\mathcal{A}$ denoteds the action space, $\mathcal{R} : \mathcal{S} \times \mathcal{A} \to \mathbb{R}$ defines the reward function, and $\mathcal{T} : \mathcal{S} \times \mathcal{A} \to \mathcal{P}(\mathcal{S})$ represents the transition function. The way an agent interacts with an MDP is determined by its policy $\pi : \mathcal{S} \to \mathcal{P}(\mathcal{A})$, which assigns distributions over actions for each state. Our main focus is on episodic MDPs, which terminate after a limited sequence of transitions. The sequence of states and actions, called a trajectory, is recorded as $z = (s_0, a_0, \ldots, s_T)$. The notation $R(z) = \sum_t \gamma^t R(s_t, a_t)$ is used to represent the total rewards accrued throughout a sequence. The primary aim of an MDP is to devise a policy that maximizes the expected cumulative reward $\mathbb{E}[R(Z)|\pi]$.

**Maximum Entropy Reinforcement Learning** a policy that exhibits a high degree of randomness is preferred in certain situations. Under these circumstances, the maximum-entropy RL objective

$$\max_{\pi} \mathbb{E}_{\pi} \left[ \sum_t R(S_t, A_t) + \beta H(A_t|S_t) \right] \tag{37}$$

serves as a compelling substitute to the conventional goal of maximizing expected aggregate rewards [39]. This objective trades off expected returns with conditional entropy of the selected policy, modulated by the temperature hyperparameter $\beta$. A generalization of the Q-value iteration method for maximum-entropy RL objective (also known as soft Bellman equation [40]) is shown in Algorithm 8.

---

**Algorithm 8** Soft Q-Value Iteration

1: **Input:** MDP, $\beta$
2: **Initialize:** $\pi_0$ to any policy
3: $i \leftarrow 0$
4: **repeat**
5: $\quad Q_{\text{soft}}^{i+1}(s, a) \leftarrow R(s, a) + \gamma \sum_{s'} \Pr(s'|s, a) \widetilde{\max}_{\beta}{}_{a'} Q_{\text{soft}}^i(s', a')$
6: $\quad i \leftarrow i + 1$
7: **until** $\|Q_{\text{soft}}^i(s, a) - Q_{\text{soft}}^{i-1}(s, a)\|_\infty \leq \epsilon$
8: **Extract policy:** $\pi_{\text{greedy}}(\cdot|s) = \text{softmax}\left(Q_{\text{soft}}^i(s, \cdot)/\beta\right)$

---

The soft maximum operator is defined as $\widetilde{\max}_{\beta}{}_a Q(s, a) = \beta \log \sum_a \exp\left(\frac{Q(s,a)}{\beta}\right)$.

## C.2 Environment Setup

For our experiments, we utilize a simple environment known as *Grid World*, for the Markov Decision Process. In this setup, the agent is placed on an $8 \times 8$ grid and, at each step, can move left, right, up, and down. The primary objective for the agent is to navigate from the starting cell to the goal cell to receive a reward of $1$, while avoiding a trap cell with a reward of $-1$. Also, the environment is noisy; even if the agent decides to move in a specific direction, the environment might, with a certain noise probability, force a move in a direction $90°$ off the intended path. The rewards received are discounted by a factor of $0.95$. Finally, the receiver has to decode a message, uniformly chosen from an alphabet of size 1024, given the final trajectory of the agent. Figure 7 illustrates the Grid World used in this experiment and depicts a trajectory taken by the agent. The environment used in the experiments is forked from the implementation of Hanselman [41].

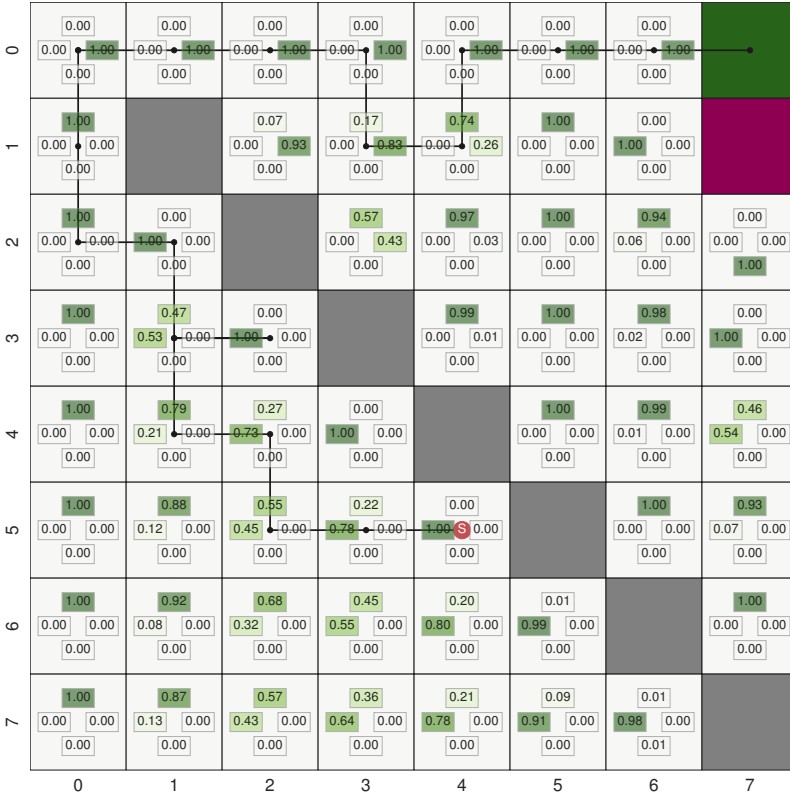

Figure 7: The Grid World Setup used in the experiments. The starting cell is depicted by a red circle, while the goal, trap, and obstacle cells are colored green, red, and grey, respectively. Additionally, a non-deterministic policy is demonstrated through the probabilities of actions in each direction within each cell. The path taken by the agent is traced in black. Note that due to the noisy environment, the agent may move in directions not explicitly suggested by the policy.

The marginal policy is learned through Soft Q-Value iteration, as described in Algorithm 8. By increasing the value of $\beta$ in Equation (14), we induce more randomness into the marginal policy. Figure 8 shows two policies learned by high and low values of $\beta$.

$\log \beta = -6$ (left table)  $\log \beta = -3$ (right table)

Figure 8: The Maximum Entropy policy learned through Soft Q-Value iteration of Algorithm 8, for $\log \beta = -6$ (left) and $\log \beta = -3$ (right).

## D   Additional Experimental Results

### D.1   Deterministic EBIM Solver vs. Shkel et al. (2017)

As discussed in Section 2, our proposed search method in Algorithm 1 is compared with the encoder from Shkel and Verdú [3]. Our formulation directly imposes an entropy constraint on the code, whereas the encoding scheme by Shkel et al. limits the code by its alphabet size. In their approach, the encoder iterates over all input symbols, assigning each one to a message that has accumulated the smallest total probability up to that point.

Figure 9 displays the mutual information obtained for each maximum allowed code rate value, considering two different input distributions. As observed, the two methods yield comparable mutual information in the high-rate regime. However, in the low-rate regime, our proposed algorithm identifies more mappings and thus significantly outperforms the encoder described in [3].

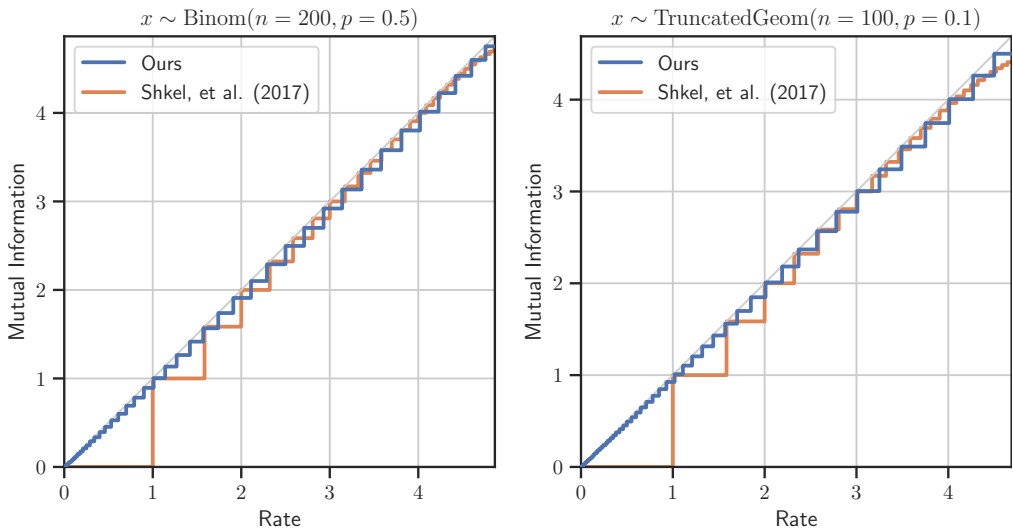

Figure 9: Obtained $I(X; T)$ vs. maximum allowed $H(T)$ for Binomial (left) and Truncated Geometric (right) input distributions.

## D.2 Visualizing Couplings from MEC-B

As discussed in Section 1, optimizing the encoder and decoder separately for the Minimum Entropy Coupling with Bottleneck (MEC-B) problem, as outlined in (2), involves first designing the encoder by solving the Entropy-Bounded Information Maximization (EBIM) in (10). This is followed by optimizing the decoder using Minimum Entropy Coupling (MEC) between the code distribution (derived from the previous step) with the output distribution.

To illustrate the couplings generated, we apply the MEC-B framework to inputs and outputs that are uniformly distributed across an alphabet of size 30. For EBIM, we only search for deterministic mappings using Algorithm 1, while for MEC, we employ the max-seeking method outlined in Algorithm 6. Figure 10 illustrates the generated couplings for varying encoder compression rates, defined by the ratio of the entropy of the input $H(X)$ to the allowed code budget $H(T)$. Greater compression rates are observed to lead to larger entropy couplings; moving from completely deterministic mappings to increasingly stochastic ones.

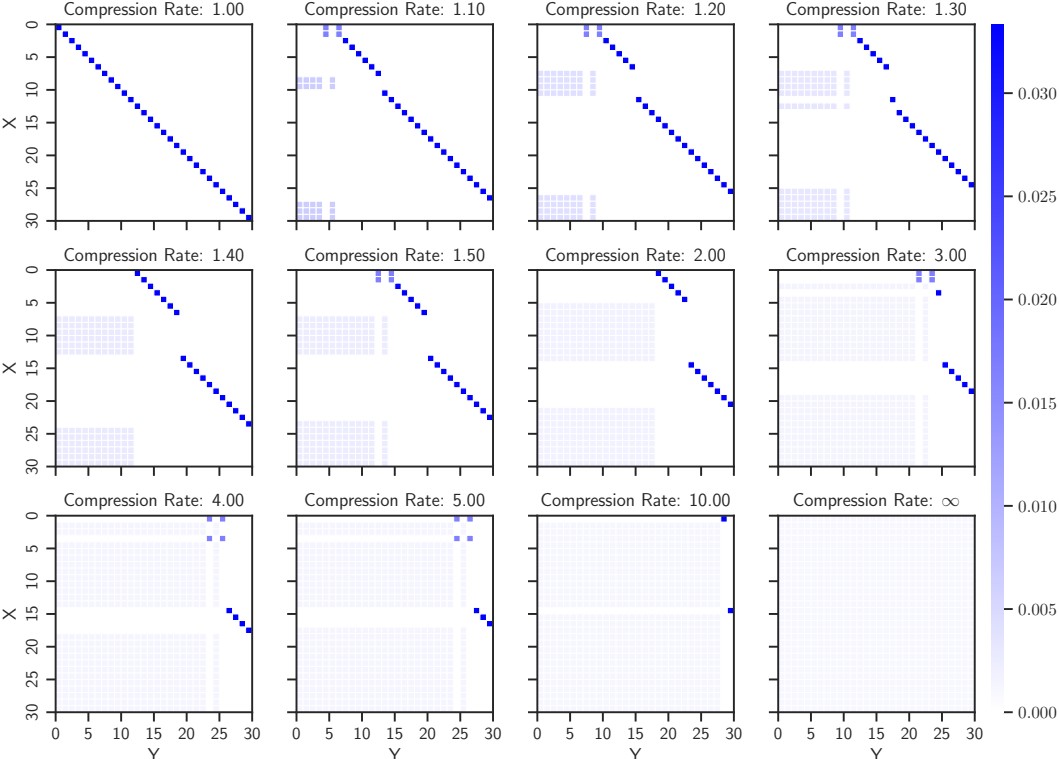

Figure 10: Generated couplings in MEC-B formulation (2), for uniform input and output distributions. The compression rate is defined as $H(X)/R$. Higher compression rates lead to more stochastic couplings with increased entropy.

# E Unsupervised Image Restoration

This section introduces a preliminary formulation and initial results for a joint image compression and upscaling task, intended to illustrate a practical application of our proposed MEC-B framework and to suggest potential avenues for future research.

We consider two unpaired datasets, $\mathcal{D}_X$ and $\mathcal{D}_Y$, which contain low-resolution and high-resolution images, respectively. Because the datasets are unpaired, there is no direct correspondence between a specific low-resolution image in $\mathcal{D}_X$ and a high-resolution image in $\mathcal{D}_Y$. In this setup, the task requires compressing a low-resolution image $X$ into a compressed representation $T$, and then reconstructing an upscaled version $Y$ from $T$.

In applying our MEC-B framework in Eq. (2), we leverage the Variational Information Maximization approach of Barber and Agakov [42]:

$$I(X;Y) = H(X) - H(X|Y) \tag{38}$$

$$= H(X) + \mathbb{E}_{y \sim p_Y} \left[ \mathbb{E}_{x \sim p_{X|Y}(\cdot|y)} \left[ \log p_{X|Y}(x|y) \right] \right] \tag{39}$$

$$= H(X) + \mathbb{E}_{y \sim p_Y} \left[ \mathbb{E}_{x \sim p_{X|Y}(\cdot|y)} \left[ \log q_\gamma(x|y) \right] + D_{\mathrm{KL}} \left( p_{X|Y}(\cdot|y) \,\|\, q_\gamma(\cdot|y) \right) \right] \tag{40}$$

$$\geq H(X) + \mathbb{E}_{y \sim p_Y} \left[ \mathbb{E}_{x \sim p_{X|Y}(\cdot|y)} \left[ \log q_\gamma(x|y) \right] \right]. \tag{41}$$

Using the Lemma 5.1 from Chen et al. [43], we have:

$$I(X;Y) \geq H(X) + \mathbb{E}_{y \sim p_Y} \left[ \mathbb{E}_{x \sim p_{X|Y}(\cdot|y)} \left[ \log q_\gamma(x|y) \right] \right] \tag{42}$$

$$= H(X) + \mathbb{E}_{x \sim p_X, \hat{y} \sim p_{Y|X}(\cdot,x)} \left[ \log q_\gamma(x|\hat{y}) \right] \tag{43}$$

In practical terms, by training the network $q_\gamma$ to model the degradation process $Y \to X$ and reconstructing $x$ from $\hat{y}$, we maximize a lower bound on the mutual information $I(X;Y)$. Simultaneously, we enforce the output distribution constraint $p_Y$ via an adversarial loss from a discriminator $d_\rho$. The total loss is therefore composed of an information loss and an adversarial loss, $\mathcal{L} = \mathcal{L}_{\mathrm{info}} + \lambda \mathcal{L}_{\mathrm{adv}}$. Figure 11 illustrates a block diagram of this framework. Note that we use a deterministic encoder $f_\theta$ that outputs the quantized code $T$, while the generator remains stochastic due to the addition of noise $z$.

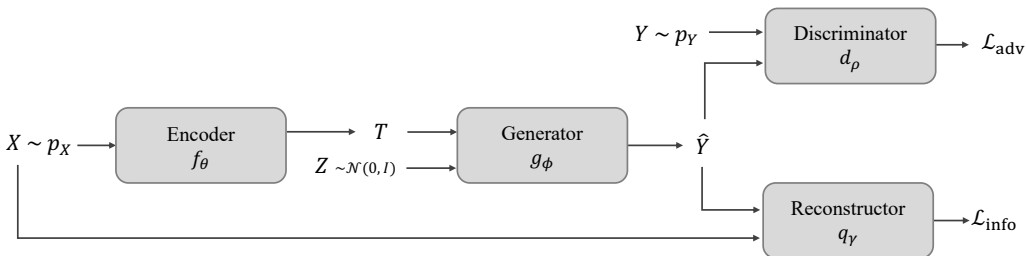

Figure 11: Block diagram of the unsupervised image restoration framework.

Figures 12 and 13 present sample output results after training the networks to achieve $4\times$-upscaling of the input images, using the MNIST [44] and SVHN [45] datasets. Observing Figure 13, we notice some color discrepancies between the original and upscaled images. This discrepancy highlights an inherent property of mutual information: for any invertible function $f$, $I(X;Y) = I(X; f(Y))$. This implies that the objective function is invariant to feature permutations, including transformations like color rotations in image channels, which can manifest as color distortions in the output. To mitigate such artifacts, careful architectural or design constraints are required to properly address feature permutations.

code bits/dim

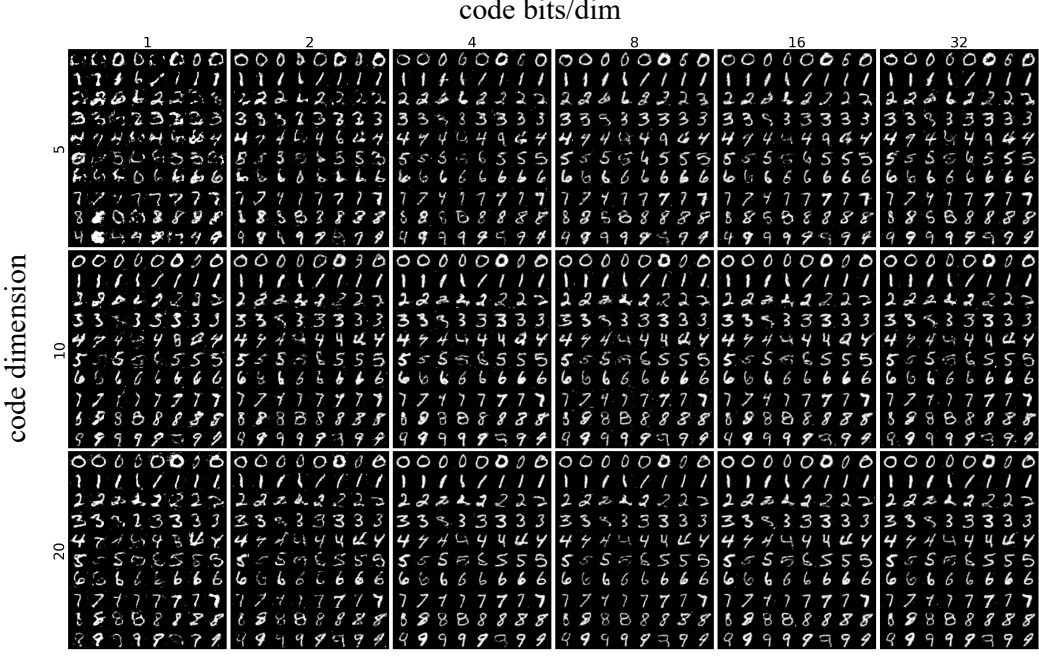

Figure 12: Output samples from the MNIST dataset, for different number of code dimensions and the number of bits per dimension of the code.

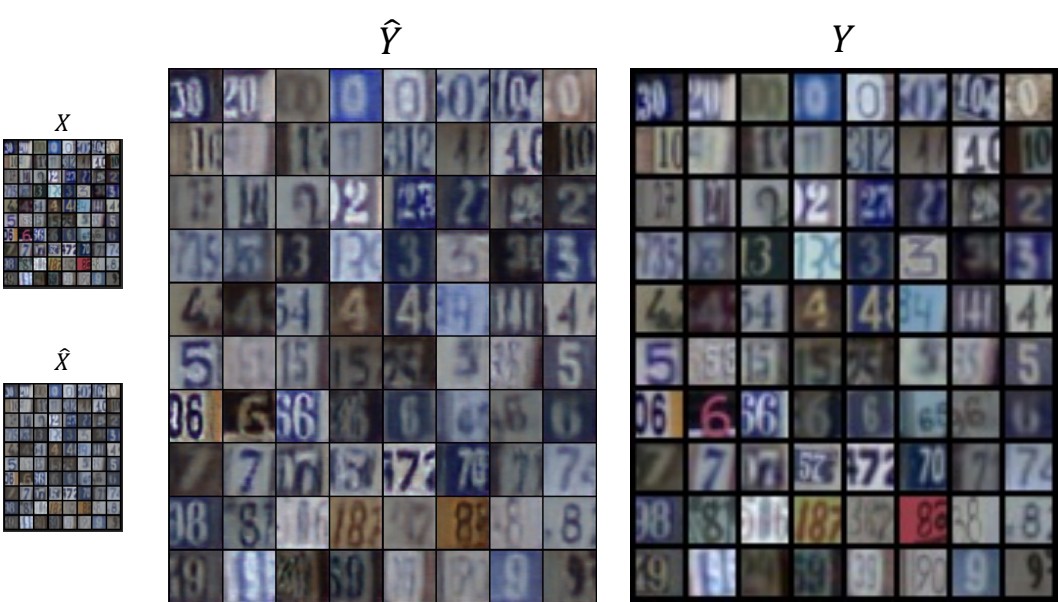

Figure 13: Input and output samples from the SVHN dataset.

