# OpenReview forum: "Minimum Entropy Coupling with Bottleneck"
_NeurIPS.cc/2024/Conference — NeurIPS 2024 spotlight_

### Official Review · Reviewer_mDET · 2024-06-24

**Soundness:** 4
**Presentation:** 3
**Contribution:** 4
**Rating:** 8
**Confidence:** 4

**Summary:**

The submission formulates and provides an approximate solution for the entropy-bounded information maximization problem. The submission extends Markov coding games to a rate limited setting and illustrates how its entropy-bounded information maximization approach can be to address such settings.

**Strengths:**

There are a number of nice contributions in the submission. The submission formulates a very natural problem (entropy-bounded information maximization) and provides a solution with provable approximation guarantees. It further shows how entropy-bounded information maximization can be used to address an interesting extension of Markov coding games to rate limits.

**Weaknesses:**

>  One of the earliest greedy algorithms for MEC was introduced by [21] in the context of causal inference, achieving a local minimum with a gap of 1 + log n bits from the optimum, where n represents the size of the alphabet.

The paper *Minimum-Entropy Coupling Approximation Guarantees Beyond the Majorization Barrier* (i.e., [6]) shows that [21]'s algorithm achieves within $log_2(e) \approx 1.44$ bits of optimal entropy for n variable coupling and $log_2(e)/e \approx 0.53$ bits of optimal entropy for 2 variable coupling, which are stronger guarantees than that stated above.

---

As a general comment, using number citations as proper nouns is bad practice: "[30] explores a single-shot lossy source coding setting
under logarithmic-loss, using a straightforward encoding scheme." This should instead be "Shkel and Verdú (2017) explore a single-shot lossy source coding setting under logarithmic-loss, using a straightforward encoding scheme."

---

The submission's Theorem 1 evokes the paper *Bounds on the Entropy of a Function of a Random Variable and their Applications* to me. It would be worth discussing as related work.

---

I was walking through Algorithm 1 and it seems not to handle the following edge case appropriately: Take the case where R = H(X). Then Algorithm 1 will return $p_{s}^{(1)}$ when we would prefer it to return $\text{diag}(p_X)$.

---

Writing:

Let pXT denoted by a |X | × |T | matrix, defines a deterministic mapping T = g(X), with I(X; T) = H(T) = Rg.

---

> 1. I_EBIM(pX, Rg + ϵ) is attained by transferring an infinitesimal probability mass from the cell with the lowest value in a column, after normalizing each column by dividing by its sum, to a new column of pXT .

Both this statement and the subsequent statement (2) should be stated more clearly. The way the statement currently reads, it's not clear why normalization changes anything (the cell with the lowest value in a column is invariant to column normalization). It would also be good to be more clear about the fact that the mass is being transferred to a column in the same row.

---

> Marginal Policy Before execution, we first derive a marginal policy π for the MDP, based on the Maximum-Entropy reinforcement learning objective

The submission should probably attribute this high level approach to [32].

The submission should probably also explain more generally how Algorithm 2, 3, 4 extend the approach of [32].

**Questions:**

> It is straightforward to check that the optimal solution of (1) is achieved when T = Y, with the identity decoder. To address this issue of decoder collapse, we introduce a constraint on the output marginal distribution, P(Y )

Is decoder collapse actually an issue? If (1) is actually the problem of interest, what makes the identity decoder undesirable?

**Limitations:**

Have the authors adequately addressed the limitations

Yes.

---

> ### Author Rebuttal · Authors · 2024-08-06
>
> >paper … [6] shows that [21]'s algorithm achieves within 𝑙𝑜𝑔2(𝑒)≈1.44  bits of optimal entropy for n variable coupling and  𝑙𝑜𝑔2(𝑒)/𝑒≈0.53  bits of optimal entropy for 2 variable coupling, which are stronger guarantees than that stated above.
>
> We thank the reviewer for pointing this out. Upon further review, it appears that in [4] this bound is improved to $\log_2(e) ≈ 1.44$ bits, while in [6] it is further refined to $\log_2(e)/e ≈ 0.53$ bits for two distributions and $(1 + \log_2(e)) / 2 ≈ 1.22$ bits for any number of distributions. We will incorporate these updates into the related work section.
>
> >As a general comment, using number citations as proper nouns is bad practice…
>
> Thank you for the feedback. We will correct this in the final version of the paper.
>
> >The submission's Theorem 1 evokes the paper Bounds on the Entropy of a Function of a Random Variable and their Applications to me. It would be worth discussing as related work.
>
> We appreciate the connection to this work and will include a discussion of this paper in the related work section of our revised manuscript.
>
> The deterministic EBIM solver in Algorithm 1 addresses a more general version of the problem discussed in the referenced work.  Specifically, while the referenced work aims at finding $\max_f H(f(X))$, the deterministic EBIM solver in Algorithm 1 provides a greedy approximation for $\max_f H(f(X))$ with $H(f(X)) \leq R$. Additionally, the NP-hardness proof provided in this work can help demonstrate the NP-hardness of the EBIM problem.
>
> >I was walking through Algorithm 1 and it seems not to handle the following edge case appropriately: Take the case where R = H(X). Then Algorithm 1 will return 𝑝𝑠(1) when we would prefer it to return diag(𝑝𝑋).
>
> Thank you. We fixed this in the manuscript by adding the following check before the main for loop:
>
> if $R \geq H(X)$ then
> $\quad$ return diag($p_X$)
>
>
> >Both this statement and the subsequent statement (2) [in thm 3] should be stated more clearly. The way the statement currently reads, it's not clear why normalization changes anything (the cell with the lowest value in a column is invariant to column normalization). It would also be good to be more clear about the fact that the mass is being transferred to a column in the same row.
>
> We have revised the statements in Theorem 3 for increased clarity:
>
> 1. $I_{EBIM} (p_X, R_g + \epsilon)$ is attained as follows:
> Normalize the columns by dividing each column by its sum. Then, select the cell with the smallest normalized value and move an infinitesimal probability mass from this cell to a new column of $p_{XT}$ in the same row.
>
> 2. $I_{EBIM}(p_X, R_g - \epsilon)$ is achieved as follows:
> Identify the columns with the smallest and largest sums in $p_{XT}$. Select the cell with the smallest value in the column with the lowest sum. Transfer an infinitesimal probability mass from this cell to the column with the highest sum in the same row.
>
> To clarify, in (1) we select the cell with the smallest column-normalized value among all cells.
>
> >”Marginal Policy Before execution, we first derive a marginal policy π for the MDP, based on the Maximum-Entropy reinforcement learning objective” The submission should probably attribute this high level approach to [32]. The submission should probably also explain more generally how Algorithm 2, 3, 4 extend the approach of [32].
>
> We updated the manuscript throughout Section 4 to clarify that the high-level approach presented in Section 4.1 follows the work of [32], where MCG is introduced. We would like to point out that we have extended the method in [32] by adding a bottleneck between the source and the agent. In [32], the agent fully observes the message, while in our extension, the source first compresses the message using the EBIM formulation, and the agent only observes this compressed message. The agent and receiver’s algorithms have been updated accordingly to encode and decode the compressed message.
>
> >Is decoder collapse actually an issue? If (1) is actually the problem of interest, what makes the identity decoder undesirable?
>
> The addition of an output distribution constraint is a practical necessity, as in a lossy compression setup the decoder needs to generate outputs following a desired distribution. For example, in image restoration, the output consists of reconstructed images from the code adhering to a certain distribution, possibly the same as the input distribution. Additionally, we would like to point out that the special case of an identity decoder is essentially the EBIM problem, which has been thoroughly investigated in the paper as a subproblem of the general framework.

---

> > ### Comment · Reviewer_mDET · 2024-08-09
> > **Response**
> >
> > Thanks for your response!
> >
> > It could be helpful to readers to give more concrete specifics about the necessity of the output distribution constraint in the text.
> >
> > ---
> >
> > Enjoyed the paper! Nice job.

---

### Official Review · Reviewer_Enxp · 2024-07-12

**Soundness:** 4
**Presentation:** 3
**Contribution:** 3
**Rating:** 7
**Confidence:** 4

**Summary:**

The paper develops a lossy compression framework that uses logarithmic loss (using conditional entropy or mutual information). This framework is designed to handle scenarios where the reconstruction distribution diverges from the source distribution, making it particularly relevant for applications requiring joint compression and retrieval under distributional shifts. The work introduces the Minimum Entropy Coupling with Bottleneck (MEC-B) framework, and considers two decomposition strategies based on fixing either the encoder (MEC) or the decoder (EBIM).  Since the general solution to the problem is intractable, the authors propose a greedy algorithm for EBIM with guaranteed performance and a study of the performance near the optimal deterministic solutions. The theoretical work is followed by a demonstration of its applicability to a Markov game setting with communication bottleneck, demonstrating its practical applicability, and quantifying the tradeoff between reward and decoding accuracy.

**Strengths:**

The general problem of lossy compression under logarithmic loss (MEC-B) is interesting, and worthy of study. Relating the quality measure of MEC-B to two related problems, MEC and EBIM, in Lemma 1, is interesting, even if not very surprising. For the EBIM problem it is shown that the optimal encoder is deterministic, an interesting result in itself. The greedy algorithm suggested is novel to the best of my knowledge, and proved to be optimal up to a constant related to the source distribution. These results, with the added characterization of Theorem 3 provide a good characterization of the problem and its space of solutions, especially in situations of distribution shift between input and output.

**Weaknesses:**

The main weakness, as noted by the authors, is that the full MEC-B problem is left open, while special cases of it have been solved. The restriction to discrete alphabet settings is also a limitation of the approach. Extending to more general cases is expected to lead to very challenging analysis.

**Questions:**

(1) To help the reader assess the quality of the gap in Theorem 2, please provide some orders of magnitude assessments for the mutual information relative to the gap in terms of the problems attributes (e.g., size of alphabet). (2) Section 4: The receiver receives a full trajectory of the MDP. This means that the algorithm is an offline MDP setting. Is this correct? (3) What are the computational costs of the algorithm proposed? (4) The notion of distributional shift is motivated in the Introductions, but received little discussion later on. Especially as Algorithm 1 is for the EBIM problem where the output distribution is only constrained in terms of entropy.

**Limitations:**

The results are limited to discrete alphabets, and sub-problems of the full MEC-B problem.  There is a recent line of work dealing with rate-perception-distortion setups (starting from the work of Blau and Michaeli titled “Rethinking Lossy Compression: The Rate-Distortion-Perception Tradeoff”, ICML 2019, with many follow-ups). While they do not use logarithmic distortions, it would be interesting to discuss this line of work in a comparative setting, since many of the problems, and particularly the tradeoffs, are similar.

---

> ### Author Rebuttal · Authors · 2024-08-06
>
> >the full MEC-B problem is left open, while special cases of it have been solved
>
> Our approach involves decomposing the full MEC-B problem into two subproblems: EBIM and MEC. By optimizing the encoder and decoder separately, we can reconstruct a solution to the full MEC-B problem. First, we optimize the encoder by solving EBIM between the input and the code. Then, given the resulting marginal distribution $p_T$​ on the code, we optimize the decoder by solving a minimum entropy coupling between the code and the output marginals. We acknowledge that joint optimization of the encoder and decoder presents an important direction for future research.
>
> >The restriction to discrete alphabet settings is also a limitation of the approach
>
> The current work focuses on the case of discrete alphabets to establish foundational grounds and provide insight into this problem. Formulating the continuous case involves additional considerations, which is a direction for future research. We will briefly discuss why the continuous setting requires a revised formulation:
>
> First, note that under the current formulation, we need to keep the code discrete. We cannot simply replace entropy with differential entropy since, in general, we will have $I(X;Y)=\infty$. On the other hand, for the case where $X$ and $Y$ are continuous but $T$ is discrete, the optimal solution can be easily found as follows:
>
> Given the rate constraint $R$, we fix any (discrete) marginal distribution on $T$ with $H(T) = R$. Next, since $X$ is continuous, a deterministic conditional $p_{T|X}$ can be trivially found by partitioning the domain of $X$ according to $p_T$, i.e., finding $(-\infty, x_1), [x_1, x_2), \cdots, [x_n, \infty)$ such that $P(X \in [x_{i-1}, x_{i}]) = P(T = i)$. (A similar argument can be inferred from Theorem 2, where in this case we have $h(p_2)=0$.) Similarly, the conditional $p_{T|Y}$ can be found in the same way, allowing $p_{Y|T}$ to be computed. It is easy to see (from Lemma 1) that $I(X; Y) = R$, which is the optimal value for the MEC-B objective.
>
> A potential approach to address these challenges is to replace information measures $I(X;Y)$ and $H(T)$ by information dimension measures so that the relevant quantities do not equal infinity. In this new formulation, compression takes the form of dimension reduction. We will include a discussion on this topic in the updated version of the paper.
>
> >(1) provide some orders of magnitude assessments for the mutual information relative to the gap...
>
> We would like to point out that the gap stated in Theorem 2 does not exceed one bit, i.e., $h(p_2) \leq 1$, with equality occurring with $p_X = [0.5, 0.5]$. While the gap is capped at one bit, the maximal mutual information in the EBIM formulation will be on the order of magnitude of $R$. Therefore, the most natural interpretation of the gap occurs in the higher rate regimes. Additionally, the gap in Theorem 2 will be small when $p_2$​ is small, i.e., in scenarios where the input alphabet size is large and the distribution is not skewed towards a few elements.
>
> >(2) The receiver receives a full trajectory of the MDP. This means that the algorithm is an offline MDP setting. Is this correct?
>
> To clarify, before the execution of the Markov Coding Game, the marginal policy is learned using Maximum-Entropy RL objective. Beyond this, there is no limitation on the type of RL used; it can be conducted either online or offline. In the execution phase, as mentioned, the receiver needs access to the full trajectory of the episode. This can be achieved offline, where the receiver obtains the full trajectory at the end of the episode, or online, where the receiver observes the current action and state at each step along with the agent, and updates its belief on the message accordingly. In this regard, we consider the MDP to be fully observable. An extension based on partially observable MDPs could be considered, though it is outside the scope of this paper.
>
> >(3) What are the computational costs of the algorithm proposed?
>
> Algorithm 1 has $\mathcal{O}(n\log n)$ time complexity, where $n$ is the cardinality of the input alphabet.
> This is because the main loop of the algorithm runs for at most $n$ steps, and finding min/max elements can be done in $\mathcal{O}(\log n)$ using a heap data structure. Also, the mutual information calculation at each step can be done in constant time by only calculating the decrease in entropy after combining two elements of the distribution.
>
> >(4) distributional shift is motivated in the Introduction but received little discussion later on...
>
> We would like to clarify that the notion of distributional shift is mathematically treated in the formulation of our main problem, MEC-B. As stated in Eq. (2), we allow input and output distributions to differ, accommodating the distributional shift. On the other hand, the proposed EBIM problem in Equation (4) is intended as a subproblem and special case of the main proposed formulation, designed to optimize the encoder within the MEC-B framework.
>
> >a recent line of work on rate-perception-distortion (Blau and Michaeli) ...
>
> We thank the reviewer for their insightful suggestion. We'd like to highlight the following points to draw a high-level comparison between the two frameworks:
>
> 1. The RDP framework of Blau and Michaeli does not fix the output distribution but instead imposes a softer perceptual constraint on the generated outputs. Additionally, our work incorporates an entropy constraint on $T$ as a rate bottleneck, whereas in Blau and Michaeli’s formulation, $I(X; Y)$ can be interpreted as the rate bottleneck.
>
> 2. In this line of work, [25] is the closest to our approach in spirit, as the authors consider a hard output distribution constraint while incorporating a bottleneck. However, that work mainly considered squared error distortion, and their mathematical machinery is quite different from our work.
>
> We will include a discussion in the revised version of the paper.

---

> > ### Comment · Reviewer_Enxp · 2024-08-10
> >
> > Thank you for your response. This is a good paper with solid theory and interesting potential uses. I have raised my score by 1 point.

---

### Official Review · Reviewer_vk4R · 2024-07-14

**Soundness:** 4
**Presentation:** 3
**Contribution:** 4
**Rating:** 8
**Confidence:** 4

**Summary:**

The authors introduce a novel lossy compression framework which introduces a bottleneck to the canonical minimum entropy coupling framework. The authors show that the encoder task can be solved approximately by a novel greedy algorithm with guaranteed performance, while the decoder problem reduced to a canonical minimum entropy coupling problem. The authors test their approach on a novel Markov Coding Game variant that admits rate limits and find that it outperforms a quantization baseline.

**Strengths:**

The paper is well-structured, well-written, and defines a new - and potentially very powerful - framework. The authors' theoretical contributions are strong; particularly the novel greedy algorithm for solving the EBIM problem is impressive. The authors deliver a solid empirical evaluation of their proposed algorithm.

**Weaknesses:**

A substantial weakness of the paper is its lack of evaluation on tasks with clear real-world applications. The authors allude to a number of possible real-world applications in the conclusion, but the MCG setting is fairly removed from real-world impact.

## Minor

* in equation (1), R is not yet defined

**Questions:**

* In terms of real-world impact, could the authors further elaborate how their EBIM algorithm would contribute to robust watermarking approaches?

**Limitations:**

The authors discuss limitations adequately.

---

> ### Author Rebuttal · Authors · 2024-08-06
>
> >in equation (1), R is not yet defined
>
> Thank you for pointing this out. We have fixed this and introduced R before presenting equation (1).
>
> >In terms of real-world impact, could the authors further elaborate how their EBIM algorithm would contribute to robust watermarking approaches?
>
> We consider the proposed formulation of MEC-B as a general setting that extends and encompasses multiple frameworks:
>
> - Minimum Entropy Coupling: The framework can be viewed as an extension of the well-known minimum entropy coupling framework, incorporating a bottleneck. This involves generating couplings with minimized entropy greater than a certain lower-bound, allowing for controlled entropy in the generated couplings.
>
> - Lossy Compression Setup: Another perspective on this framework is a general lossy compression setup with distributional shift and log-loss as the distortion measure. From this viewpoint, the framework can encompass applications such as joint compression and retrieval.
>
> - Markov Coding Games (MCG): We discussed an extension of  Markov coding games with bottleneck, which involves sequentially embedding a coded message in the choice of actions of an agent interacting with an MDP. MCG is a general framework that includes scenarios like referential games.
>
> - Watermarking: Similar to Markov Coding Games, where a message is embedded sequentially in the actions of an agent, watermarking involves embedding a message in tokens sampled autoregressively from a language model. Thus another potential application is watermarking LLM-generated text by generating couplings between the output token distribution of the language model and the information message.

---

> > ### Comment · Reviewer_vk4R · 2024-08-12
> > **Thanks for Your Response**
> >
> > I thank the authors for their response.
> >
> > While I agree that the authors discuss various real-world applications, I would not consider any of them as moderate or high impact.
> >
> > I struggle to see how MCGs would be useful in the real world, and using the authors' approach in watermarking would run into severe robustness issues. The most exciting avenue for real-world application seems to be that of lossy compression; however here the authors haven't supplied any convincing empirical evidence for any high impact settings (say e.g. image compression vs, JPEG).
> >
> > Given I am already advocating for acceptance, I will maintain my current score. For a higher score, the authors would need to demonstrate more tangible real-world impact.

---

> > > ### Author Response · Authors · 2024-08-14
> > > **Thank you for your comment!**
> > >
> > > Dear Reviewer,
> > >
> > > Thank you for your comments and the constructive feedback on our work.
> > >
> > > Our primary goal with this paper was to introduce and lay the foundational groundwork for this general problem and its potential applications. We note that broader applicability might require additional problem-specific adaptations, which are beyond the scope of this initial study.
> > >
> > > To address the practical application aspect, we are working on including preliminary formulation and results for a joint image compression and upscaling task in the appendix of our paper. This inclusion is intended to illustrate a practical application and suggest possible future research directions based on our framework.
> > >
> > > We appreciate your feedback and hope that this addition addresses some of your concerns.

---

> > > > ### Comment · Reviewer_vk4R · 2024-08-14
> > > > **Thanks for Addressing my Concern**
> > > >
> > > > Given the authors' willingness to test their intriguing method on a more tangible application domain, I have decided to increase my score.

---

### Author Rebuttal · Authors · 2024-08-07

We would like to thank the reviewers for their positive evaluation of our paper. We appreciate their thoughtful and constructive feedback. We have addressed their comments in detail in the responses provided below.

---

### Decision · Program_Chairs · 2024-09-25

**Decision:**

Accept (spotlight)

**Comment:**

The paper presents an interesting lossy compression framework under log loss, Minimum Entropy Coupling with Bottleneck. The reviewers all agree with the contribution and novelty of the paper. Please include the promised changes in the camera ready version.